# Exploiting the Asymmetric Uncertainty Structure of Pre-trained VLMs on the Unit Hypersphere

**Li Ju**[*†]   **Max Andersson**[†]   **Stina Fredriksson**[†]   **Edward Glöckner**[†]

**Andreas Hellander**[†]   **Ekta Vats**[†]   **Prashant Singh**[†‡]

## Abstract

Vision-language models (VLMs) as foundation models have significantly enhanced performance across a wide range of visual and textual tasks, without requiring large-scale training from scratch for downstream tasks. However, these deterministic VLMs fail to capture the inherent ambiguity and uncertainty in natural language and visual data. Recent probabilistic post-hoc adaptation methods address this by mapping deterministic embeddings onto probability distributions; however, existing approaches do not account for the asymmetric uncertainty structure of the modalities, and the constraint that meaningful deterministic embeddings reside on a unit hypersphere, potentially leading to suboptimal performance. In this paper, we address the asymmetric uncertainty structure inherent in textual and visual data, and propose AsymVLM to build probabilistic embeddings from pre-trained VLMs on the unit hypersphere, enabling uncertainty quantification. We validate the effectiveness of the probabilistic embeddings on established benchmarks, and present comprehensive ablation studies demonstrating the inherent nature of asymmetry in the uncertainty structure of textual and visual data.

## 1   Introduction

Vision-language models (VLMs) have demonstrated impressive capabilities in understanding visual and linguistic information, by constructing a joint embedding space that aligns image and text embeddings [4, 29]. Models like CLIP [18] and BLIP [14], enable a wide range of downstream applications, such as zero-shot classification [1], image-to-text retrieval [5], optical character recognition [27]. However, due to the inherent ambiguity within text and image data, aligned point estimates for text and image embeddings produced by deterministic VLMs may not fully capture the complex relationships between the visual and linguistic embedding spaces.

Alternatively, instead of learning deterministic point estimate embeddings, probabilistic embedding methods [8] map text and language data onto probability distributions, more effectively capturing the ambiguity and uncertainty of the data. However, they entail training probabilistic VLMs from scratch on massive datasets – an expensive process that forgoes the strengths of established deterministic VLMs. To explore post-hoc methods for constructing probabilistic embeddings from pretrained deterministic VLMs, recent frameworks such as ProbVLM [25] and BayesVLM [3] have been proposed. However, the abmiguous nature of text-to-image retrieval is due to both textual abstraction and image variability, while image-to-text ambiguity arises from the multiplicity of textual descriptions. Existing methods overlook this asymmetric nature in texts and images, and the fact that the embeddings of most pre-trained VLMs reside on a unit hypersphere, instead of the Euclidean space.

---

[*]Corresponding author (`li.ju@it.uu.se`)

[†]Department of Information Technology, Uppsala University, Uppsala, Sweden

[‡]Science for Life Laboratory, Uppsala University, Uppsala, Sweden

39th Conference on Neural Information Processing Systems (NeurIPS 2025).

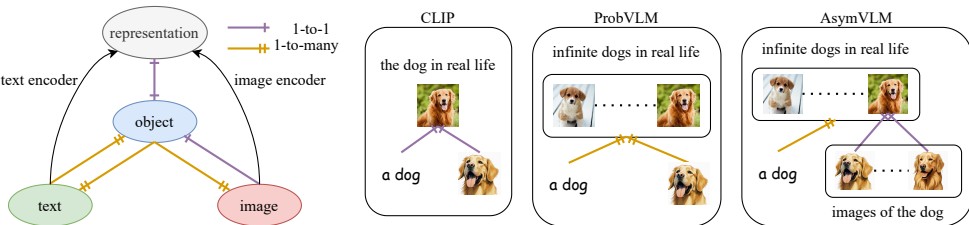

Figure 1: **Left**: Mapping relations between text, image, object, and representation spaces. **Right**: Comparison of methods – CLIP uses deterministic one-to-one mappings; ProbVLM introduces one-to-many mappings via probabilistic embeddings; AsymVLM further captures asymmetry structure between texts, images and objects.

In this work, we propose a post-hoc method, Asymmetric Probabilistic Vision-Language Model (AsymVLM), to model the uncertainty of embeddings obtained from pretrained VLMs

- We formally address the asymmetric uncertainty structure inherent in vision–language data, high aleatoric uncertainty in text versus lower aleatoric uncertainty in images.
- We propose AsymVLM, an adapter that exploits the asymmetric uncertainty structure and performs post-hoc probabilistic adaptation on the unit hypersphere, rather than in Euclidean space. We also show that AsymVLM is a natural extension of CLIP's cosine-similarity loss with the added ability to quantify uncertainty.
- Empirically, AsymVLM yields more accurate uncertainty estimates and higher cross-modal retrieval accuracy on multiple benchmarks. We further demonstrate its advantages in robust fine-tuning, zero-shot classification and "none-of-the-above" rejection.

We formulate the problem, addressing the asymmetric structure of vision-language data in Section 3, and derive our method, AsymVLM, in Section 4. We then present empirical evaluations, including uncertainty quantification, ablation studies, and downstream applications in Section 5. We conclude and discuss future work in Section 6.

## 2 Related work

**Vision-language models**   Recent advancements in vision-language modeling have facilitated the development of multimodal models capable of learning a shared embedding space for images and text. These models typically achieve this by optimizing directional similarity metrics, such as cosine similarity. Notable works including CLIP [18], BLIP [14], SigLIP [28], typically align paired image-text representations by maximizing the cosine similarity between corresponding embeddings. This strategy has proven to be effective for large-scale training, enabling these models to capture semantic relationships, and generalize well across diverse downstream tasks. However, these pretrained VLMs offer deterministic mappings that fail to capture the inherent ambiguity in the inputs.

**Probabilistic VLMs**   Methods have been proposed to improve deterministic VLM embeddings by learning a unified probabilistic embedding space for both visual and textual data. Methods such as PCME [8], and PFE [24] learn to map texts and images to aligned probability distributions, by maximizing the cross-modal likelihood between corresponding text and image data. However, these approaches require training from scratch, which is computationally expensive as existing high-quality deterministic VLMs are not fully leveraged. A more resource-efficient alternative is the probabilistic adaptation of pretrained VLMs. ProbVLM [25] trains an adapter to construct probability distributions from deterministic embeddings, while BayesVLM [3] estimates the uncertainty of cosine similarity through post-hoc Bayesian posterior approximation. As mentioned in Section 1 (and detailed in the following Section), although having demonstrated their efficacy for uncertainty quantification, existing post-hoc approaches do not consider the asymmetry in uncertainty structures between textual and visual data, and the constraint that meaningful embeddings of most pretrained VLMs reside on the unit sphere, resulting in sub-optimal performance.

**Embeddings on the unit hypersphere**   Cosine similarity is widely utilized in contemporary VLMs and foundation models pre-trained with contrastive loss [6, 18, 7]. This approach inherently constrains the learned embeddings to reside on the unit hypersphere. These embeddings have been shown to

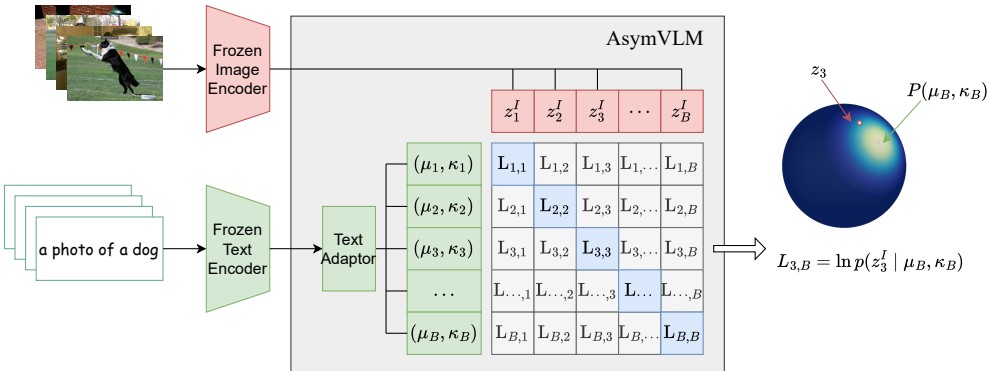

Figure 2: Asymmetric probabilistic adaptation (AsymVLM): Texts are encoded with a frozen text encoder and adaptor to produce probabilistic hyperspherical embeddings (e.g., vMF distribution), while image are deterministically encoded via a frozen image encoder. The log likelihood matrix $L$, with element $L_{m,n}$ representing the log likelihood of image vector $z_n^I$ given text embedding $z_m^T \sim P(\mu_m, \kappa_m)$, is optimized using InfoNCE to maximize diagonals and minimize off-diagonals.

have advantages such as improved uniformity and alignment compared to their Euclidean counterparts [26]. Beyond cosine similarity, embeddings on the unit hypersphere can also be explicitly modeled by assuming directional distributions, such as the von Mises-Fisher (vMF) distribution [10]. Scott et al. [21] introduced a contrastive loss based on the vMF distribution as a pretraining method, while [20] proposes a method to distill pretrained VLMs, adapting student models with vMF assumptions for zero-shot classification. However, while these approaches focus on improving empirical performance in downstream tasks, none explicitly model the uncertainty of embeddings from pretrained VLMs.

## 3 Asymmetric structure of texts and images

The relationship between vision and language exhibits an inherent asymmetry in its uncertainty structure. A textual concept, such as "dog," acts as an abstract type that can map to a vast number of unique real-world instances (tokens). Each of these instances can, in turn, be captured in countless images differing in viewpoint, lighting, and composition. This cascaded one-to-many mapping—from abstract type to physical token, and from token to specific image—induces high aleatoric uncertainty in the text modality.

Conversely, an image portrays a single, specific token. While it depicts one object unambiguously, it can be described by multiple valid captions (e.g., "a dog," "a golden retriever," "a pet"). This one-to-many mapping, from image to text, arises from linguistic diversity. Therefore, the uncertainty in text-to-image retrieval (rooted in abstraction and visual variability) is fundamentally different from that in image-to-text retrieval (rooted in descriptive variance). Despite this structural asymmetry as shown in Figure 1, existing post-hoc probabilistic adapters typically treat both retrieval directions symmetrically.

### 3.1 Mathematical formulation

Let $\mathcal{T}, \mathcal{I}, \mathcal{O}$ and $\widetilde{\mathcal{O}}$ denote the text, image, (real-world) object and (numerical) representation space respectively. The relationships between $t \in \mathcal{T}, i \in \mathcal{I}, o \in \mathcal{O}$ and $\tilde{o} \in \widetilde{\mathcal{O}}$ can be presented as follows.

**Building VLMs** Existing datasets such as MS-COCO [15] and Flickr30K [17] collect data in the form of $\{(t_n, i_n)\}_{n=1}^N$ such that there exist $o_n \in \mathcal{O}$ which satisfies $o_n \in \Phi_T(t_n)$ and $i_n \in \Phi_I(o_n)$. Based on such datasets, we aim to find a $d$-dimensional real-valued representation space $\widetilde{\mathcal{O}} \subset \mathbb{R}^d$, such that there exists a bijective function $\psi : \mathcal{O} \to \widetilde{\mathcal{O}}$. More importantly, the goal is to approximate the composite function $\psi \circ \Phi_T : \mathcal{T} \to \mathcal{P}(\widetilde{\mathcal{O}})$ and $\psi \circ \Phi_I^{-1} : \mathcal{I} \to \widetilde{\mathcal{O}}$ with finite samples from datasets as the text encoder $f_T$ and image encoder $f_T$, following the Platonic representation hypothesis [11].

| MAPPING | DEFINITION / DERIVATION | ONE-TO-MANY? |
|---|---|---|
| Text → Object | $\Phi_T(t) := \{o \in \mathcal{O} : o \text{ can be described by } t\}$ | ✓ |
| Object → Image | $\Phi_I(o) := \{i \in \mathcal{I} : i \text{ is an image of } o\}$ | ✓ |
| Text → Image | $(\Phi_I \circ \Phi_T)(t) = \bigcup_{o \in \Phi_T(t)} \Phi_I(o)$ | ✓ |
| Image → Object | $\Phi_I^{-1}(i) := o$ | ✗(assumption) |
| Object → Text | $\Phi_T^{-1}(o) := \{t \in \mathcal{T} : o \text{ can be described by } t\}$ | ✓ |
| Image → Text | $(\Phi_T^{-1} \circ \Phi_I^{-1})(i) = \{t \in \mathcal{T} : \Phi_I^{-1}(i) \in \Phi_T(t)\}$ | ✓ |
| Object ↔ Representation | $\psi(o) := \tilde{o}, \ \psi^{-1}(\tilde{o}) = o$ | ✗ |
| Text encoder | $f_T \approx \psi \circ \Phi_T$ | ✓ |
| Image encoder | $f_I \approx \psi \circ \Phi_I^{-1}$ | ✗ |

## 3.2 Motivating the approach

Trained on datasets in the form of $\{(t_n, i_n)\}_{n=1}^N$, existing deterministic pre-trained models maximize the similarity measure between matching text-image pairs while minimizing it for non-matching pairs [18, 28]. These approaches effectively learn a real-valued embedding space $\widetilde{\mathcal{O}}$, along with deterministic text and image encoders. Existing probabilistic adaptation methods, rather than mapping an image or caption to a single point in the embedding space, represent elements of $\mathcal{T}$ or $\mathcal{I}$ as random variables in $\widetilde{\mathcal{O}}$, thereby capturing the inherent one-to-many nature of the relationships.

However, the relationships among texts, images, and objects are inherently asymmetric, leading to distinct structures in text-to-image and image-to-text retrieval tasks. While both tasks are one-to-many, the asymmetry arises from different sources: abstraction and visual variability in text-to-image versus diverse textual descriptions in image-to-text. This structural difference motivates the use of asymmetric probabilistic structure for encoders within the representation space.

*From an uncertainty-quantification perspective*, we adopt the standard taxonomy of aleatoric vs. epistemic uncertainty [12]. Aleatoric uncertainty captures irreducible data ambiguity, e.g. the caption "a dog" denotes an entire class of breeds, poses and contexts, so no amount of additional training can pinpoint a single underlying object. Epistemic uncertainty captures model ignorance that could be reduced with more or better-distributed data, e.g., images, while depicting a unique scene with low aleatoric spread, may still suffer from coverage gaps if certain viewpoints or lighting conditions were under-represented during training. This asymmetry yields inherently broad aleatoric uncertainty in text versus more data-dependent epistemic uncertainty in images. In practice, aleatoric uncertainty is modeled via parameterized distributions, whereas epistemic uncertainty is estimated by Monte Carlo methods (e.g., sampling from dropout layers or Bayesian neural networks).

Assuming an underlying asymmetric uncertainty structure, a direction for improvement emerges. For text embeddings, it is essential to capture the wide variance inherent in language via distributions to reflect the multitude of corresponding objects. Conversely, image embeddings can be be modeled deterministically and the uncertainty is reduced through large coverage of visual data in the datasets.

## 4 Method

With a pretrained VLM such as CLIP or SigLIP, we denote the text encoder as $f_T : \mathcal{T} \to \mathbb{S}^{d-1}$ and image encoder as $f_I : \mathcal{I} \to \mathbb{S}^{d-1}$, where $d$ is the dimension of the representation space.

### 4.1 Asymmetric probabilistic adaptation

As discussed in Section 3, capturing the variance is crucial for text embeddings, while ensuring coverage in training data is key for image embeddings. To achieve this, we introduce a text adapter $g_T$ for the text encoder $f_T$, while retaining the deterministic embeddings from the image encoder $f_I$, as expanding image data coverage lies beyond the scope of this work. Formally, the embedding of any text $t \in \mathcal{T}$ is modeled by a random variable $\boldsymbol{z}^T$,

$$\boldsymbol{z}^T \sim P(\theta(t)) \text{ where } \theta(t) := g_T \circ f_T(t), \tag{1}$$

where $P(\theta(t))$ denotes a distribution parametrized by $\theta(t)$ predicted by a neural network given corresponding text. The deterministic image embedding $z^I$ from the image encoder $f_I$ is retained.

**Objective function** For any batch of data $\{(t_1, i_1), \ldots, (t_B, i_B)\}$, we aim to maximize $\ln p_{\boldsymbol{z_n^T}}(z_n^I)$ for all $n \in [B]$ while minimizing $\ln p_{\boldsymbol{z_m^T}}(z_n^I)$ for all $n, m \in [B]$, $n \neq m$, where $p_{\boldsymbol{a}}(b)$ denotes the value of the PDF of random variable $\boldsymbol{a}$ at $b$. First we define the log likelihood matrix $L \in \mathbb{R}^{B \times B}$, where the $(m, n)$-th element is computed as $L(m, n) = \ln p_{\boldsymbol{z_m^T}}(z_n^I)$, for $n, m \in [B]$.

To maximize the diagonal elements and minimize the off-diagonal elements of $L$, we apply the InfoNCE loss to $L$ and obtain the objective function to be minimized,

$$\underset{\theta \in \Theta}{\arg\min} - \frac{1}{2B} \sum_{n=1}^{B} \left[ \ln \frac{\exp\left(\tau \cdot L(n, n)\right)}{\sum_{m=1}^{B} \exp\left(\tau \cdot L(n, m)\right)} + \ln \frac{\exp\left(\tau \cdot L(n, n)\right)}{\sum_{m=1}^{B} \exp\left(\tau \cdot L(m, n)\right)} \right], \quad (2)$$

where $\tau$ is a temperature parameter and $\theta$ is the parameter set of distribution $P_{\boldsymbol{z}^T}(\theta(t))$.

## 4.2 Modelling on the unit hypersphere

The shared representation space built by most pre-trained VLMs is $\mathbb{S}^{d-1}$, the domain in which both $\boldsymbol{z}^T$ and $z^I$ reside. This observation necessitates the selection of a directional distribution for $P(\theta(t))$, defined on $\mathbb{S}^{d-1}$. To properly construct a probabilistic embedding on $\mathbb{S}^{d-1}$, directional distributions for the embeddings should be employed – either projected distributions (e.g., the angular normal distribution) or native directional distributions (e.g., the vMF distribution). In this work, we use the vMF, and power spherical (PS) distributions to model the text embeddings distribution. The former is generally considered as the simplest directional distribution, while the latter is reported to be an alternative to the vMF distribution with a closed form normalization constant.

**Derivation w.r.t. vMF distribution** The vMF distribution $\mathrm{vMF}(\mu, \kappa)$ is a probability distribution on $\mathbb{S}^{d-1}$, parameterized by a mean direction $\mu \in \mathbb{R}^d$ and an isotropic concentration parameter $\kappa \in \mathbb{R}$, where $\|\mu\|_2 = 1$ and $\kappa \geq 0$. The PDF for random unit vector $x$ is given by,

$$p(x; \mu, \kappa) = C_d(\kappa) \exp(\kappa \cdot \mu^\top x), \text{ where } C_d(\kappa) = \frac{\kappa^{d/2-1}}{(2\pi)^{d/2} I_{d/2-1}(\kappa)},$$

and $I_p$ denotes the modified Bessel function of the first kind, at order $p$.

Following Equation 1, assuming that $\boldsymbol{z}^T \sim \mathrm{vMF}(\mu(t), \kappa(t))$ for text $t \in \mathcal{T}$, for any image $i \in \mathcal{I}$, we have $\ln p_{\boldsymbol{z}^T}(z^I) = \kappa \cdot \mu(t)^\top z^I + \ln C_d(\kappa(t))$. However, computing $C_d(\kappa)$ is intractable in the high dimensional space, yet [21] have shown that $\ln C_d(\kappa)$ can be approximated by $F_d(\kappa) + \text{const.}$, where $F_d(\kappa)$ is given by,

$$F_d(\kappa) = \frac{d-1}{4} \ln \left( \frac{d-1}{2} + \sqrt{\left(\frac{d-1}{2}\right)^2 + \kappa^2} \right) - \frac{1}{2}\sqrt{\left(\frac{d-1}{2}\right)^2 + \kappa^2}$$

$$+ \frac{d-1}{4} \ln \left( \frac{d-1}{2} + \sqrt{\left(\frac{d+1}{2}\right)^2 + \kappa^2} \right) - \frac{1}{2}\sqrt{\left(\frac{d+1}{2}\right)^2 + \kappa^2}.$$

Injecting the log likelihood of vMF into Equation 2 results in the vMF kernel for the objective function $L_{\mathrm{vMF}}$ as follows,

$$L_{\mathrm{vMF}}(r, s) = \kappa(t_r) \cdot \mu(t_r)^\top z_s^I + F_d(\kappa(t_r)), \; \forall r, s \in [B].$$

The detailed derivation for the approximation is deferred to Appendix A.1.

**Derivation w.r.t. PS distribution** The power spheric distribution $\mathrm{PS}(\mu, \kappa)$ is an alternative to vMF, also parametrized by a mean direction $\mu \in \mathbb{R}^d$ and a concentration parameter $\kappa \in \mathbb{R}$ with $\|\mu\|_2 = 1$ and $\kappa \geq 0$. The PDF of a PS distribution is given by,

$$p(x; \mu, \kappa) = (1 + \mu^\top x)^\kappa \cdot C_d(\kappa), \text{ where } C_d(\kappa) = \left\{ 2^{\alpha+\beta} \pi^\beta \frac{\Gamma(\alpha)}{\Gamma(\alpha + \beta)} \right\}^{-1},$$

and $\alpha = (d-1)/2 + \kappa, \beta = (d-1)/2$.

Similarly, following Equation 1, assuming that $\boldsymbol{z}^t \sim \mathrm{PS}(\mu(t), \kappa(t))$ for text $t \in \mathcal{T}$, for any image $i \in \mathcal{I}$, injecting the log likelihood of PS distribution in Equation 2 results in the PS kernel $L_{\mathrm{PS}}$,

$$
\begin{aligned}
L_{\mathrm{PS}}(r,s) = {}& \kappa(t_r)\ln(1 + \mu(t_r)^\top z_s^I) - (d - 1 + \kappa(t_r))\ln 2 \\
& - \ln\Gamma\left(\frac{d-1}{2} + \kappa(t_r)\right) + \ln\Gamma(d - 1 + \kappa(t_r)), \ \forall r, s \in [B].
\end{aligned}
$$

### 4.3 Uncertainty quantification

We quantify text-embedding uncertainty as $u(t) = 1/\kappa(t)$, due to the fact that for both the vMF and PS distributions on $\mathbb{S}^{d-1}$, the (angular) variance $\sigma^2$ is a strictly decreasing function of $\kappa$. Equivalently, larger $\kappa$ concentrates mass more tightly around the mean direction (lower $\sigma^2$), so $1/\kappa$ grows monotonically with the distribution's variance, providing a natural scalar value for uncertainty estimate for probabilistic embeddings.

### 4.4 Connections to CLIP

Equation 2 can be rewritten in the following form,

$$
\theta = \underset{\theta \in \Theta}{\arg\min} -\frac{1}{2B}\sum_{n=1}^{B}\left[\ln\frac{\exp\left(\tau\delta(n,n)\right)}{\sum_{m=1}^{B}\exp\left(\tau\delta(n,m)\right)} + \ln\frac{\exp\left(\tau\delta(n,n)\right)}{\sum_{m=1}^{B}\exp\left(\tau\delta(m,n)\right)}\right].
$$

Denoting $\mathrm{CosSim}(r,s) = \mu(t_r)^\top z_s^I$, for any $r, s \in [B]$ we have,

$$
\begin{aligned}
&\text{for CLIP: } \delta_{\mathrm{CLIP}}(r,s) = \mathrm{CosSim}(r,s), \\
&\text{for AsymVLM}_{\mathrm{vMF}}: \delta_{\mathrm{vMF}}(r,s) = \kappa(t_r)\cdot\mathrm{CosSim}(r,s) + F_d(\kappa(t_r)), \qquad (3)\\
&\text{for AsymVLM}_{\mathrm{PS}}: \delta_{\mathrm{PS}}(r,s) = \kappa(t_r)\ln(1 + \mathrm{CosSim}(r,s) + \ln C_d(\kappa(t_r)).
\end{aligned}
$$

Given that $\kappa(t_r) \geq 0$, it is clear that the latter two objectives are monotonically increasing w.r.t. $\delta_{\mathrm{CLIP}}$ with an additional parameter $\kappa$. These can be interpreted as *extensions of the CLIP loss by introducing $\kappa$, the concentration parameter, to model the variance of the text embedding without drifting from the pre-trained models using cosine similarities.*

> Equation 3 demonstrates AsymVLM's consistency with the underlying ideas of CLIP, while incorporating mechanisms to account for embedding variance. AsymVLM ultimately offers a more refined and probabilistically grounded approach to modeling semantic relationships between text and images. Furthermore, AsymVLM can be extended to a SigLIP variant. Inference using AsymVLM also extends beyond simply computing the cosine similarity, to utilizing the maximum likelihood principle, with a clearer statistical explanation. Details about the inference process and its SigLIP-variant are provided in Appendix A.2 and A.3.

## 5 Empirical results

We compare AsymVLM to the baselines on benchmark datasets, and subsequently present an ablation study. Further, extended analyses and demonstration on two downstream applications are presented.

**Datasets, baselines and metrics** The MS-COCO [15] and Flickr-30k [17] datasets are used to train the adapters. Additionally, a subset of the Conceptual Caption dataset [23] of 200k samples (CC-200k) is also used. To understand the nature of learned uncertainty, the HierarCap dataset [2] is used, containing four captions of different abstraction levels for each image. The baselines include BayesVLM, ProbVLM, ProLIP, PFE and PCME++ adapted from pretraining methods (denoted by PFE* and PCME++*, respectively). The pre-trained VLM used in the experiments in the main text is CLIP (ViT-B/32 backbone). We assess the uncertainty quality in text embeddings by evaluating recall@1 performance in cross-modal retrieval tasks. A strong positive correlation between uncertainty and errors indicates that higher uncertainty leads to poorer embeddings and lower recall. For each task, we group results by uncertainty levels and compute Spearman's rank correlation ($S$) between uncertainty and recall [22]. We then fit a regressor ($R^2$) to determine if the

performance drop is linear, and compare average Recall@1 across uncertainty levels for different methods. All experiments were repeated *five times with different random seeds*, and we report the mean results with full standard deviations provided in the Appendix. The code for this paper is available at https://github.com/li-ju666/asymvlm.

## 5.1 Evaluation of UQ and cross-modal retrieval

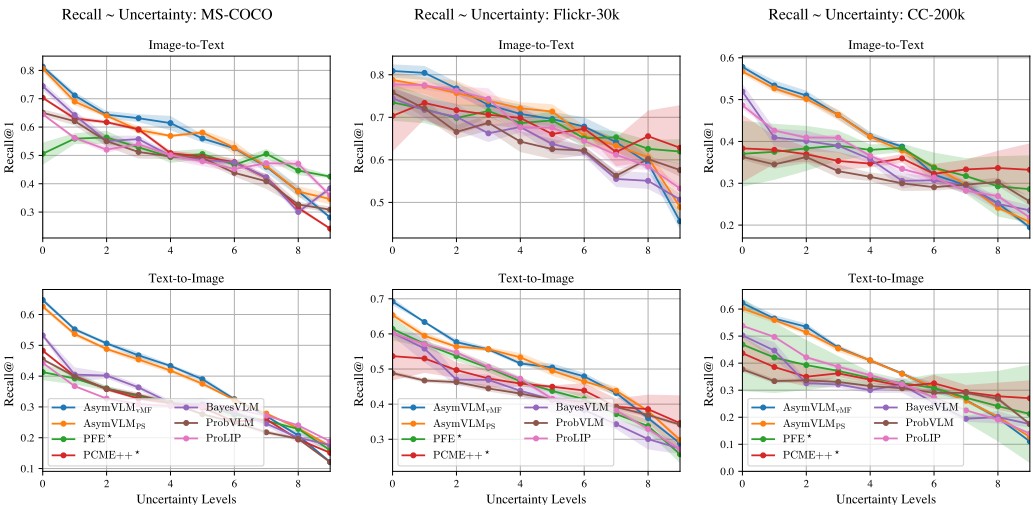

Figure 3: Recall@1 versus uncertainty levels for cross-modal retrieval tasks for various datasets. Each subplot shows the performance degradation as uncertainty increases for PFE⋆, PCME++⋆, ProbVLM, AsymVLM$_{vMF}$, and AsymVLM$_{PS}$. Ideally Recall@1 should decrease monotonically.

Table 1: Evaluation of model performance and uncertainty on benchmarks (MS-COCO, Flickr-30k, CC-200k) for Image-to-Text (i2t) and Text-to-Image (t2i) retrieval. We report Recall@1 ↑, regression fit $R^2$ ↑ between uncertainty and retrieval error, and Spearman's rank $S$ ↓ correlation between uncertainty and retrieval error. **Bold** font denotes the best results and underline denotes the second best results. Each value is the mean of 5 runs with standard deviation in the Appendix, Table A.2.

| | | T2I | | | I2T | | |
|---|---|---|---|---|---|---|---|
| **DATASET** | **METHOD** | Recall@1 ↑ | $R^2$ ↑ | $S$ ↓ | Recall@1 ↑ | $R^2$ ↑ | $S$ ↓ |
| **MS-COCO** | PFE⋆ | 0.500 | 0.558 | -0.722 | 0.304 | 0.946 | -0.988 |
| | PCME++⋆ | 0.500 | 0.931 | **-0.996** | 0.304 | 0.948 | -0.990 |
| | BayesVLM | 0.506 | 0.884 | -0.976 | 0.323 | 0.932 | -0.985 |
| | ProbVLM | 0.480 | **0.951** | -0.981 | 0.293 | 0.979 | **-1.000** |
| | ProLip | 0.500 | 0.808 | -0.908 | 0.304 | 0.876 | -0.985 |
| | AsymVLM$_{vMF}$ | **0.561** | 0.948 | -0.988 | **0.392** | 0.984 | **-1.000** |
| | AsymVLM$_{PS}$ | 0.558 | 0.937 | -0.984 | 0.390 | **0.989** | **-1.000** |
| **FLICKR-30K** | PFE⋆ | 0.680 | 0.675 | -0.832 | 0.451 | 0.955 | **-0.998** |
| | PCME++⋆ | 0.679 | 0.455 | -0.178 | 0.451 | 0.900 | -0.918 |
| | BayesVLM | 0.637 | **0.916** | **-0.976** | 0.425 | 0.934 | -0.973 |
| | ProbVLM | 0.646 | 0.826 | -0.914 | 0.422 | **0.964** | -0.985 |
| | ProLip | 0.678 | 0.829 | -0.954 | 0.450 | 0.928 | -0.978 |
| | AsymVLM$_{vMF}$ | **0.688** | 0.860 | **-0.976** | **0.504** | 0.960 | -0.995 |
| | AsymVLM$_{PS}$ | **0.688** | 0.854 | -0.947 | 0.498 | 0.946 | -0.993 |
| **CC-200K** | PFE⋆ | 0.352 | 0.857 | -0.521 | 0.336 | 0.988 | -0.595 |
| | PCME++⋆ | 0.352 | 0.198 | -0.148 | 0.336 | 0.779 | -0.887 |
| | BayesVLM | 0.347 | 0.905 | -0.968 | 0.304 | 0.874 | -0.937 |
| | ProbVLM | 0.316 | 0.772 | -0.837 | 0.302 | 0.775 | -0.965 |
| | ProLip | 0.351 | 0.968 | -0.990 | 0.335 | 0.992 | **-1.000** |
| | AsymVLM$_{vMF}$ | **0.395** | 0.990 | **-1.000** | **0.383** | 0.991 | -0.998 |
| | AsymVLM$_{PS}$ | 0.393 | **0.992** | **-1.000** | 0.380 | **0.993** | **-1.000** |

Figure 3 and Table 1 summarize the main results comparing the proposed AsymVLM with baseline methods. For uncertainty quantification, results indicate that while all methods capture text uncertainty to some degree, AsymVLM consistently outperforms its counterparts. As the uncertainty level

increases, AsymVLM is the only method for which Recall@1 decreases strictly monotonically. Moreover, AsymVLM also meets or exceeds competing approaches on both $R^2$ and $S$ scores.

In cross-modal retrieval, AsymVLM$_{\text{vMF}}$ and AsymVLM$_{\text{PS}}$ achieve the highest and second-highest Recall@1 scores across all datasets. Note that the retrieval performances of PFE$^\star$ and PCME++$^\star$ are identical, as they only model the variance of embeddings while while keeping embedding directions constant, resulting in performance identical to the pre-trained models. In contrast, both AsymVLM and ProbVLM adapt the directions and learn the uncertainty of the embeddings during post-hoc training. Although ProbVLM achieves reasonable uncertainty estimates overall, it consistently under-performs pre-trained models in recall, whereas AsymVLM delivers richer uncertainty estimates and improves cross-modal retrieval performance.

Furthermore, the similar performance of AsymVLM$_{\text{vMF}}$ and AsymVLM$_{\text{PS}}$ in both cross-modal retrieval and uncertainty quantification shows the robustness of AsymVLM w.r.t. the choice of the spherical distribution, suggesting possibilities of exploring other spherical or directional distributions.

## 5.2 Ablation study

We conduct an ablation study on the two important components of AsymVLM: the asymmetric architecture of the adapter, and the use of a spherical distribution for modeling. To assess the impact of two components, we designed following adapters with experimental results presented in Figure 4:

- **AsymVLM(image):** This variant modifies AsymVLM by only mapping deterministic image embeddings to spherical distributions, leaving text embeddings deterministic.

- **SymVLM:** This variant map both deterministic image and text embeddings to spherical distributions, with InfoNCE applied to symmetrized log likelihood kernel.

- **AsymVLM($\mathcal{G}$):** This variant has identical architecture with AsymVLM but maps deterministic text embedding on Gaussian distribution, followed by normalization. The kernel of the loss function is replaced by Gaussian log likelihood.

**Asymmetric architecture**  Both AsymVLM(image) and SymVLM perform worse than the original AsymVLM with respect to uncertainty quantification. Specifically, AsymVLM(image) struggles to model image uncertainty, showing no decrease in Recall@1 as uncertainty increases. Although SymVLM exhibits a slight decrease in Recall@1 with rising uncertainty, uncertainty estimates remain sub-optimal to AsymVLM. AsymVLM(image) achieves average Recall@1 comparable to AsymVLM while SymVLM performs much worse than either AsymVLM or AsymVLM(image). The results underscore the importance of an asymmetric architecture for uncertainty quantification and cross-modal retrieval performance.

**Choice of spherical distributions**  For both Image-to-Text and Text-to-Image retrieval tasks, AsymVLM($\mathcal{G}$) yields reasonable uncertainty estimates. As uncertainty levels increase, the Recall@1 of AsymVLM($\mathcal{G}$) declines almost linearly, exhibiting behaviour similar to AsymVLM. However, the overall Recall@1 is inferior compared to AsymVLM. This emphasizes not only the effectiveness of the asymmetric architecture for uncertainty quantification, but also the necessity of spherical distributions for the post-hoc adaptation of pre-trained VLMs.

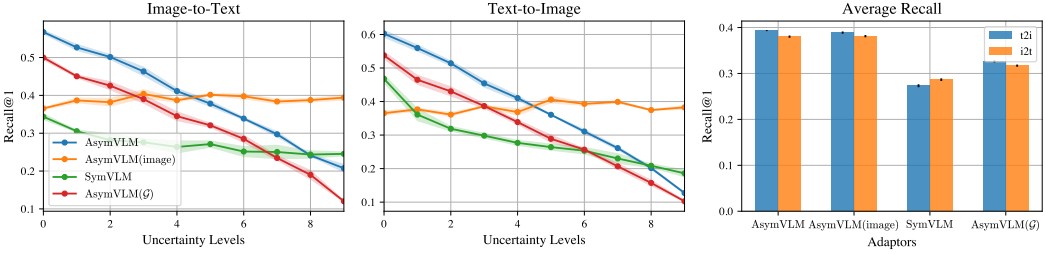

Figure 4: Ablation study results for AsymVLM: The left and middle panel shows Recall@1 performance for cross-modal retrieval across various uncertainty levels. The right panel summarizes average Recall@1 for both retrieval tasks.

## 5.3 Understanding the uncertainty estimates

To better understand the uncertainty estimates produced by AsymVLM, we analyze caption estimates on the HierarCap dataset. In this dataset, each image is paired with hierarchical captions spanning four levels – from general descriptions (Level 0) to detailed image descriptions (Level 3). Figure 5 illustrates the following observations.

- **AsymVLM captures hierarchical caption structure** Level 0 captions with general descriptions exhibit higher uncertainty estimates, indicating greater ambiguity, while Level 3 captions with the most detail generally show lower uncertainty. This pattern aligns with the inherent ambiguity of language and the corresponding uncertainty estimates provided by AsymVLM.
- **Detailed captions enhance image retrieval likelihood** The likelihood of the image embedding increases as more detailed descriptions are provided, as shown by the shift towards higher log likelihood values from Level 0 to 3. This suggests that detailed captions better capture the nuances of the image, facilitating more accurate image retrieval.
- **Longer captions exhibit reduced ambiguity** The results indicate that a higher token count is associated with lower uncertainty in the text, as depicted by the downward trend in uncertainty values. This is aligned with the expectation, as shorter texts generally tend to be more ambiguous, while longer captions provide more context, reducing ambiguity.

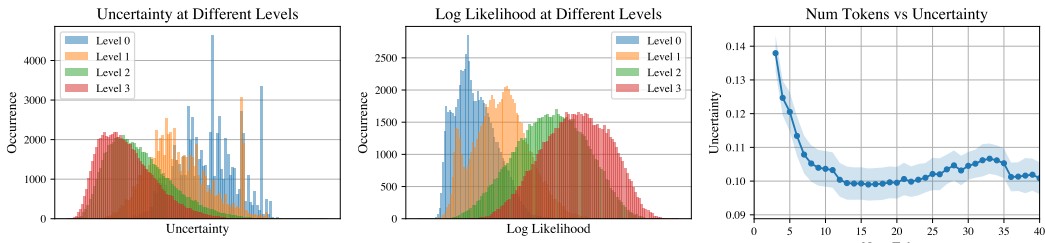

Figure 5: Uncertainty and likelihood analysis with the HierarCap dataset. The left panel shows the distribution of uncertainty estimates across hierarchical caption levels, showing decreasing uncertainty from Level 0 to 3. The middle panel illustrates the log likelihood distribution of image embeddings, with increased likelihood as caption detail increases. The right panel shows the relationship between the number of tokens and uncertainty, indicating reduced uncertainty with longer captions.

## 5.4 Applications

Below, we present two example applications of the uncertainty estimates obtained from AsymVLM, demonstrating their potential use cases.

**Handling none-of-the-above** For zero-shot classification, strategies to enable a model to say `"none-of-the-above"` rather than assigning an incorrect label include threshold-based rejection, margin-based rejection and other techniques. AsymVLM provides an alternative approach, by introducing a dummy prompt such as `"a photo"` to signal `"none-of-the-above"`. We evaluate these methods on a combined test set consisting of CIFAR-10 and CIFAR-100 samples, performing zero-shot classification over CIFAR-10 classes. Ideally, CIFAR-10 samples (positive samples) should be correctly classified into their respective categories, while CIFAR-100 samples (negative samples) should be rejected as `"none-of-the-above"`. AsymVLM consistently outperforms other methods (Table 2), achieving higher accuracy on both positive and negative samples.

Table 2: None-of-the-above-aware zero-shot classification using dummy prompts. The table reports the accuracy on positive samples and negative samples.

| DUMMY CLASS | METHOD | POS. ACC. | NEG. ACC. |
|---|---|---|---|
| `"a photo"` | CLIP | **0.888** | 0.009 |
| | Determ. FT | 0.769 | 0.115 |
| | AsymVLM$_{PS}$ | 0.845 | 0.547 |
| | AsymVLM$_{vMF}$ | 0.857 | **0.587** |
| `"a photo of an object"` | CLIP | **0.888** | 0.009 |
| | Determ. FT | 0.778 | 0.037 |
| | AsymVLM$_{PS}$ | 0.849 | **0.609** |
| | AsymVLM$_{vMF}$ | 0.858 | 0.557 |
| `"a photo of something"` | CLIP | **0.888** | 0.009 |
| | Determ. FT | 0.769 | 0.125 |
| | AsymVLM$_{PS}$ | 0.843 | **0.595** |
| | AsymVLM$_{vMF}$ | 0.857 | 0.585 |
| NONE | Margin-Based | 0.584 | 0.579 |
| | Threshold-Based | 0.646 | 0.560 |

Table 3: Zero-shot classification accuracy. "None" denotes the baseline without fine-tuning.

| FINE-TUNED | METHOD | CIFAR-10 | CIFAR-100 | STL-10 | IMAGENET-1K |
|---|---|---|---|---|---|
| | | **VALIDATED ON** | | | |
| **MS-COCO** | Determ. FT | 0.684 | 0.300 | 0.829 | 0.395 |
| | AsymVLM$_{PS}$ | **0.847** | 0.470 | **0.952** | 0.502 |
| | AsymVLM$_{vMF}$ | 0.837 | **0.477** | 0.940 | **0.507** |
| **FLICKR-30K** | Determ. FT | 0.688 | 0.342 | 0.874 | 0.369 |
| | AsymVLM$_{PS}$ | 0.791 | 0.413 | **0.920** | 0.451 |
| | AsymVLM$_{vMF}$ | **0.792** | **0.422** | 0.918 | **0.464** |
| **CC-200K** | Determ. FT | 0.779 | 0.411 | 0.922 | 0.450 |
| | AsymVLM$_{PS}$ | 0.861 | **0.542** | **0.968** | 0.520 |
| | AsymVLM$_{vMF}$ | **0.866** | **0.542** | 0.967 | **0.527** |
| **NONE** | CLIP | 0.888 | 0.642 | 0.974 | 0.632 |

**Robust fine-tuning** Pre-trained VLMs are renowned for their zero-shot classification performance. However, fine-tuning these models on smaller datasets typically impairs this capability on other datasets. Table 3 compares AsymVLM with its deterministic counterpart, denoted by Determ.FT, and the foundation model without fine-tuning. Models fine-tuned with AsymVLM consistently outperform those fine-tuned deterministically across all settings. This suggests that AsymVLM preserves the zero-shot classification proficiency acquired during pre-training to a greater extent, whereas the deterministic approach is more prone to catastrophic forgetting. Therefore, AsymVLM offers a more robust fine-tuning strategy for VLMs.

# 6 Conclusion and limitation

We introduced AsymVLM, a post-hoc probabilistic adaptation framework for pretrained vision–language models that leverages asymmetric uncertainty in textual and visual embeddings. Text embeddings are modeled as random variables using directional distributions (e.g., von Mises–Fisher and power spherical) on the unit hypersphere, while image embeddings remain deterministic. Incorporating these spherical likelihoods into an InfoNCE-based objective enables effective modeling of wide-variance aleatoric uncertainty in language.

While AsymVLM delivers impressive performance in cross-modal retrieval with reliable uncertainty estimates, it only models text uncertainty. Future work could extend uncertainty modeling to image embeddings via data augmentation or similar methods. Also, although example applications of the uncertainty estimates are provided, further exploration into additional downstream tasks is warranted.

## Broader impact

Uncertainty estimates for text are vital in high-stakes applications involving VLMs, such as medical diagnostics where ambiguous language must be treated carefully. AsymVLM enhances trust in textual inputs, enabling safer and more robust applications of VLMs in uncertainty-sensitive domains.

## Acknowledgement

The computations/data handling were enabled by the Berzelius resource provided by the Knut and Alice Wallenberg Foundation at the National Supercomputer Centre, and by the Alvis resource provided by Chalmers e-Commons at Chalmers. LJ acknowledges funding from the Centre for Interdisciplinary Mathematics at Uppsala University and NAISS through Project 2024/22-1358. AH and PS acknowledge support from the Swedish Research Council through grant agreement nos. 2023-05167 and 2023-05593 respectively. EV's research was partially supported by the Kjell and Märta Beijer Foundation, and the Wallenberg AI, Autonomous Systems and Software Program (WASP) funded by the Knut and Alice Wallenberg Foundation. PS also acknowledges support from the the Knut and Alice Wallenberg foundation through the Program for Academic leaders in Life Science (PALS). The authors thank Aleksandr Karakulev, Csongor Horváth and Mayank Nautiyal for valuable discussions and feedback.

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

## A   Method Details

### A.1   Approximation of the log normalizer of vMF distribution

The normalizer of vMF distribution $C_d(\kappa)$ is given by,

$$C_d(\kappa) = \frac{\kappa^{d/2-1}}{(2\pi)^{d/2} I_{d/2-1}(\kappa)}.$$

---

**Remark 1** (Derivatives of modified Bessel functions [16]). *Let $I_v(z)$ denote modified Bessel function of $v$-th order. For $k = 0, 1, 2, \ldots,$*

$$\left(\frac{1}{z}\frac{d}{dz}\right)^k (z^v I_v(z)) = z^{v-k} I_{v-k}(z).$$

---

**Lemma 1.** *The first order derivate of $I_v(z)$ is,*

$$\frac{d}{dz} I_v(z) = I_{v+1}(z) + \frac{v}{z} I_v(z).$$

*Proof.* Using Remark 1, let $k = 1$ and we have,

$$\frac{1}{z}\frac{d}{dz} z^v I_v(z) = z^{v-1} I_{v-1}(z),$$

$$v z^{v-1} I_v(z) + z^v \frac{dI_v(z)}{dz} = z^v I_{v-1}(z).$$

Reorganizing the equation gives us,

$$\frac{d}{dz} I_v(z) = I_{v-1}(z) - \frac{v}{z} I_v(z).$$

Further we use the Recurrence Relations of modified Bessel function [16],

$$I_{v-1}(z) - I_{v+1}(z) = \frac{2v}{z} I_v(z),$$

and obtain,

$$\frac{d}{dz} I_v(z) = I_{v+1}(z) + \frac{v}{z} I_v(z).$$

$\square$

---

Note that,

$$\frac{d}{d\kappa} \ln C_d(\kappa) = \frac{d}{d\kappa} \left[ (d/2 - 1) \ln \kappa - d/2 \ln 2\pi - \ln I_{d/2-1}(\kappa) \right],$$

$$= \frac{d/2 - 1}{\kappa} - \frac{dI_{d/2-1}(\kappa)/d\kappa}{I_{d/2-1}(\kappa)}.$$

Using Lemma 1, we have,

$$\frac{d}{d\kappa} \ln C_d(\kappa) = \frac{d/2 - 1}{\kappa} - \frac{I_{d/2}(\kappa) + \frac{d/2-1}{\kappa} I_{d/2-1}(\kappa)}{I_{d/2-1}(\kappa)},$$

$$= -\frac{I_{d/2}(\kappa)}{I_{d/2-1}(\kappa)}.$$

Ruiz-Antolín and Segura [19] give a tight lower and upper bound for $-\frac{I_{d/2}(\kappa)}{I_{d/2-1}(\kappa)}$, given by following Remark:

> **Remark 2** (Tight lower and upper bound for the ratio of modified Bessel functions [19])**.** *Let $I_v(z)$ be the modified Bessel function of the first kind at order $v$. We have following bound,*
>
> $$\frac{z}{v - 1/2 + \sqrt{(v + 1/2)^2 + z^2}} < \frac{I_v(z)}{I_{v-1}(z)} < \frac{z}{v - 1/2 + \sqrt{(v - 1/2)^2 + z^2}}, v \geq 1/2.$$

With the tight lower and upper bounds, $\frac{I_v(z)}{I_{v-1}(z)}$ can be approximated by,

$$\frac{I_v(z)}{I_{v-1}(z)} \approx \frac{1}{2}(g(z) + h(z)),$$

where $g(z)$ and $h(z)$ denote the left and right hand sides of the inequality.

To further approximate $\ln C_d(\kappa)$, by plugging $v = \frac{d}{2}$, we further integrate the function w.r.t $\kappa$, and obtain following approximates,

$$
\begin{aligned}
\ln C_d(\kappa) \approx & \frac{d-1}{4} \ln \left( \frac{d-1}{2} + \sqrt{\left(\frac{d-1}{2}\right)^2 + \kappa^2} \right) - \frac{1}{2}\sqrt{\left(\frac{d-1}{2}\right)^2 + \kappa^2} \\
& + \frac{d-1}{4} \ln \left( \frac{d-1}{2} + \sqrt{\left(\frac{d+1}{2}\right)^2 + \kappa^2} \right) - \frac{1}{2}\sqrt{\left(\frac{d+1}{2}\right)^2 + \kappa^2} + \text{const.}
\end{aligned}
\tag{4}
$$

### A.2   Inference with maximum log likelihood

The inference of deterministic pre-trained VLMs (i.e. cross-modal retrieval) relies on computing the cosine similarities of each possible pairs of given texts and images, while the inference for AsymVLM has a clear statistical interpretation as maximizing log likelihood w.r.t. different image-text pairs. For Image-to-Text retrieval, given a set of text embeddings $\{\boldsymbol{z}_r^T | r \in [R]\}$, where $\boldsymbol{z}_r^T \sim \text{vMF}(\mu(t_r), \kappa(t_r))$ or $\boldsymbol{z}_r^T \sim \text{PS}(\mu(t_r), \kappa(t_r))$, and an image embeddings $z^I$, the predictor is given by,

$$\hat{r} = \arg\max_{r \in [R]} \ln p_{\boldsymbol{z}_r^T}(z^I) = \begin{cases} \kappa(t_r) \cdot (\mu(t_r))^\top z^I + F_d(\kappa(t_r)) & \text{for vMF,} \\ \kappa(t_r) \ln(1 + (\mu(t_r)))^\top z^I + \ln C_d(\kappa(t_r)) & \text{for PS.} \end{cases}$$

Similarly, for Text-to-Image retrieval, given a set of image embeddings $\{z_s^I | s \in [S]\}$ and a text embedding $\boldsymbol{z}^T \sim \text{vMF}(\mu, \kappa)$ or $\boldsymbol{z}^T \sim \text{PS}(\mu, \kappa)$, the predictor is given by,

$$\hat{s} = \arg\max_{s \in [S]} \ln p_{\boldsymbol{z}^T}(z_s^I) = \begin{cases} \kappa \cdot \mu^\top z_s^I + F_d(\kappa) & \text{for vMF,} \\ \kappa \ln(1 + \mu^\top z_s^I) + \ln C_d(\kappa) & \text{for PS.} \end{cases}$$

### A.3   Derivation for the SigLIP variant

The objective function of SigLIP is defined as,

$$-\frac{1}{B} \sum_{r=1}^{B} \sum_{r=1}^{B} \ln \frac{1}{1 + \exp(\sigma(r,s)(-\tau \cdot \text{CosSim}(r,s) + b))},$$

where $\sigma(r,s) = 1$ if $r = s$ and $\sigma(r,s) = -1$ if $r \neq s$. SigLIP shares the identical "kernel", i.e. cosine similarity, with CLIP, while our methods employs different kernels with variance modeling derived from vMF/PS assumptions for the text embeddings. By replacing the kernel of the objective function of SigLIP, we can also obtain the SigLIP variant of our method.

## B   Experimental Protocol

### B.1   Adapter architecture

The text adapter is implemented as a four-layer perceptron that takes the output of a pre-trained text encoder as its input. It consists of two hidden layers, each with 1024 dimensions and activated by the

ReLU function, followed by an output layer that has the same dimensionality as the input, and no activation function. The un-normalized output of the final layer, denoted as $z_t'$, is decomposed into $z_t' = \kappa \mu_t$, where the vector $\|\mu_t\| = 1$ and the scalar $\kappa > 0$, serving as the parameters for either vMF or PS distribution. The temperature parameter in the InfoNCE objective function is trainable.

Other adapters (ProbVLM, PFE$^\star$ and PCME++$^\star$) share identical architectures as three-layer perceptron, with different number of outputs for the last layer due to the inherent different parameterization in the methods.

## B.2 Datasets

We use the datasets MS-COCO and Flickr-30k to train and validate the adapters since they provide well-annotated image–caption pairs. Additionally, we randomly sample 200k image-caption pairs from Conceptual Caption dataset [23], building a dataset (CC-200k) for the training and validation. For ablation studies, all methods are trained on the training set of CC-200k and evaluated on the validation set of CC-200k. For the study of understanding the learned uncertainty, the model is evaluated on HierarCap dataset [2], a dataset adapted from Conceptual Caption to reflect the hierarchical structure of captions. For each image, there exists four captions of different abstraction levels from general to detailed ones. For example applications, CIFAR-10, CIFAR-100 [13] and STL10 [9] are used for zero-shot classification.

**Metrics**  The evaluation of the quality of aleatoric uncertainty / ambiguity estimates for text embeddings is based on the recall performance in cross-modal retrieval tasks. A strong positive correlation between uncertainty and error indicates that when the model is more uncertain, it tends to produce lower-quality embeddings, which leads to poorer retrieval performance. For each task, we first group the recall results according to the uncertainty levels of the text embeddings. Then we compare Spearman's rank correlation ($S$) between the uncertainty levels and recall for cross-modal retrieval [22]. We also compute the regression fit $R^2$ between the uncertainty levels and Recall@1 performances to measure if the drop in performance follows a linear trend. To evaluate the overall performance of cross-modal retrieval of the models trained with different methods, we also compare the average Recall@1 across all uncertainty levels.

## B.3 Baseline methods

The uncertainty quantification baselines include ProbVLM, PFE and PCME++ adapted/trained in a post-hoc manner from pretrained models (denoted by PFE$^\star$ and PCME++$^\star$, respectively). Note that the post-hoc adaptation for both PFE and PCME++ only predicts the variances of probabilistic embeddings, while keeping the means unchanged from the pretrained models. This is due to PFE and PCME++ achieving poor performance (worse than pre-trained models) for cross-modal retrieval, when used to to adapt the means of the embeddings. For downstream tasks, vanilla pre-trained vision-language models are also included. BayesVLM is not compared against with other methods in the main experiments, as it does not provide a single scalar value quantifying the uncertainty of its embeddings, rather scalar uncertainty for cosine similarities propagated by probabilistic embeddings.

## B.4 Optimization protocol

All methods are implemented in `PyTorch`, and all pre-trained VLMs are loaded by `transformers`. All computations are conducted on NVIDIA A100/A40 GPUs. We use stochastic gradient descent with momentum (SGD-momen.) for the optimization of AsymVLM, PCME and PFE, and use AdamW for ProbVLM, following its official implementation. The learning rates for different methods are optimized using a grid search within $\{10^{-4}, 5 \times 10^{-4}, 10^{-3}, 5 \times 10^{-3}, 10^{-2}, 5 \times 10^{-2}\}$, and reported as Table A.1. We apply cosine annealing for learning rate scheduling with a minimal learning rate $10^{-6}$.

## B.5 Computational costs

The overhead of post-hoc adaptation with AsymVLM is negligible compared to pre-training CLIP or SigLIP. This efficiency stems from two factors: (1) the adaptor is a lightweight three-layer MLP operating in the 512-dimensional latent space (for CLIP); (2) all CLIP embeddings for adaptation

Table A.1: Optimizer configurations for different methods

| Method | Optimizer | Learning rate | Batch size |
|--------|-----------|---------------|------------|
| AsymVLM | SGD | $10^{-2}$ | 2048 |
| ProbVLM | AdamW | $10^{-4}$ | 2048 |
| PFE$^\star$ | SGD | $10^{-3}$ | 2048 |
| PCME++$^\star$ | SGD | $5 \times 10^{-4}$ | 2048 |

data are cached after a single forward pass through the pretrained VLM. Adaptation training then proceeds solely on these cached embeddings, avoiding repeated runs through the large model and dramatically reducing compute cost. On an NVIDIA A100, completing 200 epochs of adaptation on either MS-COCO, Flickr-30K or CC-200k requires under one GPU-hour.

## C Supplementary Empirical Results

Complete experimental results for the evaluation of AsymVLM on CLIP embeddings are presented in Table A.2 (mean $\pm$ std. dev. over 5 runs).

Table A.2: Complete experimental results for the comparison of different methods for CLIP embeddings.

| DATASET | METHOD | I2T | | | T2I | | |
|---------|--------|-----|-----|-----|-----|-----|-----|
| | | Recall@1 ↑ | $R^2$ ↑ | $S$ ↓ | Recall@1 ↑ | $R^2$ ↑ | $S$ ↓ |
| **MS-COCO** | PFE$^\star$ | 0.500 ± 0.000 | 0.558 ± 0.253 | -0.722 ± 0.193 | 0.304 ± 0.000 | 0.946 ± 0.021 | -0.988 ± 0.008 |
| | PCME++$^\star$ | 0.500 ± 0.000 | 0.931 ± 0.006 | **-0.996 ± 0.004** | 0.304 ± 0.000 | 0.948 ± 0.010 | -0.990 ± 0.004 |
| | BayesVLm | 0.506 ± 0.007 | 0.884 ± 0.020 | -0.976 ± 0.011 | 0.323 ± 0.006 | 0.932 ± 0.023 | -0.985 ± 0.014 |
| | ProbVLM | 0.480 ± 0.004 | **0.951 ± 0.011** | -0.981 ± 0.006 | 0.293 ± 0.001 | 0.979 ± 0.002 | **-1.000 ± 0.000** |
| | ProLip | 0.500 ± 0.000 | 0.808 ± 0.025 | -0.908 ± 0.031 | 0.304 ± 0.000 | 0.876 ± 0.007 | -0.985 ± 0.012 |
| | AsymVLM$_{vMF}$ | **0.561 ± 0.002** | 0.948 ± 0.016 | -0.988 ± 0.013 | **0.392 ± 0.001** | 0.984 ± 0.004 | **-1.000 ± 0.000** |
| | AsymVLM$_{PS}$ | 0.558 ± 0.004 | 0.937 ± 0.013 | -0.984 ± 0.012 | 0.390 ± 0.002 | **0.989 ± 0.001** | **-1.000 ± 0.000** |
| **FLICKR-30K** | PFE$^\star$ | 0.680 ± 0.000 | 0.675 ± 0.186 | -0.832 ± 0.094 | 0.451 ± 0.000 | 0.955 ± 0.002 | **-0.998 ± 0.005** |
| | PCME++$^\star$ | 0.679 ± 0.001 | 0.455 ± 0.386 | -0.178 ± 0.650 | 0.451 ± 0.000 | 0.900 ± 0.063 | -0.918 ± 0.067 |
| | BayesVLm | 0.637 ± 0.012 | **0.916 ± 0.035** | -0.976 ± 0.013 | 0.425 ± 0.010 | 0.934 ± 0.011 | -0.973 ± 0.019 |
| | ProbVLM | 0.646 ± 0.006 | 0.826 ± 0.062 | -0.914 ± 0.029 | 0.422 ± 0.002 | **0.964 ± 0.016** | -0.985 ± 0.012 |
| | ProLip | 0.678 ± 0.000 | 0.829 ± 0.032 | -0.954 ± 0.041 | 0.450 ± 0.000 | 0.928 ± 0.010 | -0.978 ± 0.005 |
| | AsymVLM$_{vMF}$ | **0.688 ± 0.005** | 0.860 ± 0.029 | -0.976 ± 0.010 | **0.504 ± 0.001** | 0.960 ± 0.006 | -0.995 ± 0.006 |
| | AsymVLM$_{PS}$ | **0.688 ± 0.005** | 0.854 ± 0.021 | -0.947 ± 0.032 | 0.498 ± 0.002 | 0.946 ± 0.021 | -0.993 ± 0.006 |
| **CC-200K** | PFE$^\star$ | 0.352 ± 0.000 | 0.857 ± 0.075 | -0.521 ± 0.757 | 0.336 ± 0.000 | 0.988 ± 0.008 | -0.595 ± 0.798 |
| | PCME++$^\star$ | 0.352 ± 0.000 | 0.198 ± 0.375 | -0.148 ± 0.430 | 0.336 ± 0.000 | 0.779 ± 0.105 | -0.887 ± 0.060 |
| | BayesVLM | 0.347 ± 0.010 | 0.905 ± 0.025 | -0.968 ± 0.016 | 0.304 ± 0.008 | 0.874 ± 0.026 | -0.937 ± 0.022 |
| | ProbVLM | 0.316 ± 0.001 | 0.772 ± 0.102 | -0.837 ± 0.066 | 0.302 ± 0.001 | 0.775 ± 0.029 | -0.965 ± 0.021 |
| | ProLip | 0.351 ± 0.000 | 0.968 ± 0.007 | -0.990 ± 0.009 | 0.335 ± 0.000 | 0.992 ± 0.002 | **-1.000 ± 0.000** |
| | AsymVLM$_{vMF}$ | **0.395 ± 0.002** | 0.990 ± 0.003 | **-1.000 ± 0.000** | **0.383 ± 0.003** | 0.991 ± 0.002 | -0.998 ± 0.005 |
| | AsymVLM$_{PS}$ | 0.393 ± 0.001 | **0.992 ± 0.004** | **-1.000 ± 0.000** | 0.380 ± 0.002 | **0.993 ± 0.001** | **-1.000 ± 0.000** |

Complete experimental results for the evaluation of AsymVLM on SigLIP embeddings are presented in Table A.3.

Complete experimental results for the evaluation of AsymVLM for robust out-of-distribution zero-shot classification are presented in Table A.4.

Complete experimental results for the evaluation of AsymVLM for zero-shot classification, with none-of-the-above handling are presented in Table A.5.

Table A.3: Full experimental results for the comparison of different methods for SigLIP embeddings.

| | | I2T | | | T2I | | |
|---|---|---|---|---|---|---|---|
| **DATASET** | **METHOD** | Recall@1 ↑ | $R^2$ ↑ | $S$ ↓ | Recall@1 ↑ | $R^2$ ↑ | $S$ ↓ |
| **MS-COCO** | PFE$^\star$ | $0.654 \pm 0.000$ | $0.940 \pm 0.010$ | $-0.990 \pm 0.005$ | $0.472 \pm 0.000$ | $\mathbf{0.996 \pm 0.001}$ | $\mathbf{-1.000 \pm 0.000}$ |
| | PCME++$^\star$ | $0.654 \pm 0.000$ | $0.021 \pm 0.016$ | $-0.101 \pm 0.020$ | $0.472 \pm 0.000$ | $0.893 \pm 0.016$ | $-0.907 \pm 0.031$ |
| | BayesVLM$^\star$ | $0.664 \pm 0.001$ | $0.541 \pm 0.022$ | $-0.780 \pm 0.036$ | $0.488 \pm 0.002$ | $0.923 \pm 0.006$ | $-0.931 \pm 0.027$ |
| | ProbVLM | $0.649 \pm 0.001$ | $0.564 \pm 0.063$ | $-0.672 \pm 0.064$ | $0.469 \pm 0.001$ | $0.935 \pm 0.007$ | $-0.998 \pm 0.005$ |
| | ProLIP | $0.678 \pm 0.001$ | $0.791 \pm 0.031$ | $-0.866 \pm 0.042$ | $0.489 \pm 0.000$ | $0.944 \pm 0.002$ | $-0.996 \pm 0.005$ |
| | AsymVLM$_{vMF}$ | $\mathbf{0.694 \pm 0.002}$ | $\mathbf{0.948 \pm 0.018}$ | $\mathbf{-0.995 \pm 0.006}$ | $\mathbf{0.502 \pm 0.000}$ | $0.986 \pm 0.001$ | $\mathbf{-1.000 \pm 0.000}$ |
| | AsymVLM$_{PS}$ | $\underline{0.691 \pm 0.002}$ | $0.929 \pm 0.021$ | $-0.993 \pm 0.015$ | $\underline{0.497 \pm 0.001}$ | $\underline{0.987 \pm 0.002}$ | $\mathbf{-1.000 \pm 0.000}$ |
| **FLICKR-30K** | PFE$^\star$ | $0.815 \pm 0.000$ | $\underline{0.840 \pm 0.023}$ | $-0.947 \pm 0.030$ | $0.638 \pm 0.000$ | $0.922 \pm 0.005$ | $\mathbf{-0.998 \pm 0.005}$ |
| | PCME++$^\star$ | $0.814 \pm 0.000$ | $0.007 \pm 0.002$ | $0.127 \pm 0.070$ | $0.638 \pm 0.000$ | $0.472 \pm 0.016$ | $-0.625 \pm 0.027$ |
| | BayesVLM$^\star$ | $0.818 \pm 0.001$ | $0.496 \pm 0.042$ | $-0.776 \pm 0.021$ | $0.642 \pm 0.001$ | $0.515 \pm 0.019$ | $-0.742 \pm 0.024$ |
| | ProbVLM | $0.807 \pm 0.002$ | $0.519 \pm 0.097$ | $-0.714 \pm 0.119$ | $0.631 \pm 0.000$ | $0.525 \pm 0.038$ | $-0.640 \pm 0.151$ |
| | ProLIP | $0.812 \pm 0.003$ | $0.820 \pm 0.064$ | $-0.879 \pm 0.090$ | $0.636 \pm 0.001$ | $0.573 \pm 0.028$ | $-0.699 \pm 0.075$ |
| | AsymVLM$_{vMF}$ | $\mathbf{0.823 \pm 0.002}$ | $0.803 \pm 0.044$ | $\mathbf{-0.967 \pm 0.010}$ | $\underline{0.647 \pm 0.001}$ | $\underline{0.934 \pm 0.010}$ | $\mathbf{-0.998 \pm 0.005}$ |
| | AsymVLM$_{PS}$ | $\underline{0.819 \pm 0.006}$ | $\mathbf{0.891 \pm 0.024}$ | $\underline{-0.959 \pm 0.012}$ | $\mathbf{0.649 \pm 0.002}$ | $\mathbf{0.949 \pm 0.008}$ | $\mathbf{-0.998 \pm 0.005}$ |
| **CC-200K** | PFE$^\star$ | $0.503 \pm 0.000$ | $0.981 \pm 0.003$ | $\mathbf{-1.000 \pm 0.000}$ | $0.484 \pm 0.000$ | $\underline{0.992 \pm 0.004}$ | $\mathbf{-1.000 \pm 0.000}$ |
| | PCME++$^\star$ | $0.502 \pm 0.000$ | $0.933 \pm 0.009$ | $-0.956 \pm 0.010$ | $0.484 \pm 0.000$ | $0.889 \pm 0.016$ | $-0.965 \pm 0.014$ |
| | BayesVLM$^\star$ | $0.512 \pm 0.001$ | $\mathbf{0.993 \pm 0.006}$ | $-0.998 \pm 0.006$ | $0.488 \pm 0.001$ | $0.981 \pm 0.010$ | $-0.992 \pm 0.008$ |
| | ProbVLM | $0.489 \pm 0.002$ | $0.674 \pm 0.081$ | $-0.817 \pm 0.071$ | $0.477 \pm 0.000$ | $0.744 \pm 0.022$ | $-0.934 \pm 0.029$ |
| | ProLIP | $0.507 \pm 0.002$ | $0.737 \pm 0.061$ | $-0.903 \pm 0.055$ | $0.487 \pm 0.001$ | $0.795 \pm 0.019$ | $-0.962 \pm 0.010$ |
| | AsymVLM$_{vMF}$ | $\mathbf{0.516 \pm 0.002}$ | $\underline{0.991 \pm 0.004}$ | $\mathbf{-1.000 \pm 0.000}$ | $\mathbf{0.496 \pm 0.002}$ | $\mathbf{0.993 \pm 0.002}$ | $\mathbf{-1.000 \pm 0.000}$ |
| | AsymVLM$_{PS}$ | $\underline{0.514 \pm 0.001}$ | $0.984 \pm 0.003$ | $\mathbf{-1.000 \pm 0.000}$ | $\underline{0.489 \pm 0.002}$ | $0.985 \pm 0.002$ | $\mathbf{-1.000 \pm 0.000}$ |

Table A.4: Full experimental results for Out-of-distribution zero-shot classification accuracy on CIFAR-10, CIFAR-100 and STL-10 for pretrained VLMs fine-tuned with different methods on MS-COCO, Flickr-30k and CC-200k datasets.

| | | **VALIDATED ON** | | | |
|---|---|---|---|---|---|
| **FINE-TUNED ON** | **METHOD** | **CIFAR-10** | **CIFAR-100** | **STL-10** | **IMAGENET-1K** |
| **MS-COCO** | Determ. FT | $0.684 \pm 0.055$ | $0.300 \pm 0.017$ | $0.829 \pm 0.030$ | $0.395 \pm 0.002$ |
| | AsymVLM$_{PS}$ | $\mathbf{0.847 \pm 0.007}$ | $\underline{0.470 \pm 0.012}$ | $\mathbf{0.952 \pm 0.005}$ | $\underline{0.502 \pm 0.002}$ |
| | AsymVLM$_{vMF}$ | $\underline{0.837 \pm 0.010}$ | $\mathbf{0.477 \pm 0.010}$ | $\underline{0.940 \pm 0.010}$ | $\mathbf{0.507 \pm 0.003}$ |
| **FLICKR-30K** | Determ. FT | $0.688 \pm 0.030$ | $0.342 \pm 0.019$ | $0.874 \pm 0.012$ | $0.369 \pm 0.007$ |
| | AsymVLM$_{PS}$ | $0.791 \pm 0.019$ | $0.413 \pm 0.009$ | $\mathbf{0.920 \pm 0.009}$ | $\underline{0.451 \pm 0.003}$ |
| | AsymVLM$_{vMF}$ | $\mathbf{0.792 \pm 0.011}$ | $\mathbf{0.422 \pm 0.014}$ | $\underline{0.918 \pm 0.006}$ | $\mathbf{0.464 \pm 0.002}$ |
| **CC-200K** | Determ. FT | $0.779 \pm 0.037$ | $0.411 \pm 0.011$ | $0.922 \pm 0.017$ | $0.450 \pm 0.004$ |
| | AsymVLM$_{PS}$ | $\underline{0.861 \pm 0.008}$ | $\mathbf{0.542 \pm 0.005}$ | $\mathbf{0.968 \pm 0.001}$ | $\underline{0.520 \pm 0.002}$ |
| | AsymVLM$_{vMF}$ | $\mathbf{0.866 \pm 0.013}$ | $\mathbf{0.542 \pm 0.013}$ | $\underline{0.967 \pm 0.004}$ | $\mathbf{0.527 \pm 0.001}$ |
| **NONE** | CLIP | $0.888$ | $0.642$ | $0.974$ | $0.632$ |

Table A.5: Complete results for the evaluation metrics for various zero-shot classification methods using different dummy prompts. The table reports the accuracy on positive samples and negative samples when classifying all inputs into CIFAR-10 classes. Baseline methods using threshold- and margin-based rejection are included for comparison.

| **DUMMY PROMPT** | **METHOD** | **POSITIVE ACC.** | **NEGATIVE ACC.** |
|---|---|---|---|
| `"a photo"` | CLIP | $\mathbf{0.888}$ | $0.009$ |
| | Determ. FT | $0.769 \pm 0.028$ | $0.115 \pm 0.138$ |
| | AsymVLM$_{PS}$ | $0.845 \pm 0.011$ | $\underline{0.547 \pm 0.106}$ |
| | AsymVLM$_{vMF}$ | $\underline{0.857 \pm 0.010}$ | $\mathbf{0.587 \pm 0.110}$ |
| `"a photo of an object"` | CLIP | $\mathbf{0.888}$ | $0.009$ |
| | Determ. FT | $0.778 \pm 0.040$ | $0.037 \pm 0.026$ |
| | AsymVLM$_{PS}$ | $0.849 \pm 0.012$ | $\mathbf{0.609 \pm 0.119}$ |
| | AsymVLM$_{vMF}$ | $\underline{0.858 \pm 0.011}$ | $\underline{0.557 \pm 0.089}$ |
| `"a photo of something"` | CLIP | $\mathbf{0.888}$ | $0.009$ |
| | Determ. FT | $0.769 \pm 0.031$ | $0.125 \pm 0.132$ |
| | AsymVLM$_{PS}$ | $0.843 \pm 0.019$ | $\mathbf{0.595 \pm 0.149}$ |
| | AsymVLM$_{vMF}$ | $\underline{0.857 \pm 0.009}$ | $\underline{0.585 \pm 0.107}$ |
| **NONE** | Margin-Based | $0.584$ | $0.579$ |
| | Threshold-Based | $0.646$ | $0.560$ |

