# OpenReview forum: "Exploiting the Asymmetric Uncertainty Structure of Pre-trained VLMs on the Unit Hypersphere"
_NeurIPS.cc/2025/Conference — NeurIPS 2025 poster_

### Official Review · Reviewer_X6GY · 2025-06-21

**Clarity:** 4
**Significance:** 4
**Originality:** 3
**Rating:** 5
**Confidence:** 3

**Summary:**

The paper studies incoprtaing uncertainty into VLMs (e.g. CLIP), in particular, the text encoder now outputs a distribution (over the sphere) rather than a deterministic vector, and the motivation is simple: A given text description could correspond to multiple images due to the underlying ambiguity of textual description. To turn the deterministic text encoder into a stochastic one, the authors attached a text adapter on the text encoder, which outputs the parameter (mean and variance) of a vMF distribution or a PS distribution, and the adapter is trained using standard contrastive loss. The authors then demonstrate a wide collection of interesting applications using the uncertainty provided by the model.

Overall I find the paper very well written, and the proposed method simple and promising. I am not an expert in SSL / VLM, but I've worked on uncertainty quantification a lot, and I think this paper should be a solid accept.

**Questions:**

- Would love to see how such probabilistic vision language encoding model can be used in multimodality LLM such as  LLaVA, to enhance uncertainty quantification.

- How is the adapter implemented, is it a linear projection head attached to the pre-trained encoder model?

- Is the uncertainty captured by AsymVLM aleatoric or epistemic? I think it should be aleatoric right?

- What could happen when the test image goes out of distribution with respect to the data used for training the adapter?

**Ethical Concerns:**

["NO or VERY MINOR ethics concerns only"]

**Limitations:**

(Minor) The method potentially requires some significant overhead training the adapter due to the large training dataset size, and since the large batch size (2048) requirement may limit the accessibility of the method. It would be better if the author could discuss the total GPU hours required.

**Quality:**

4

**Strengths And Weaknesses:**

# Strengths

- The paper is very well motivated.

- The paper is very well organized and well written, and the figures are clear, illustrative, and informative.

- The empirical results are very thorough and interesting! The experiment considers a wide range of settings related to uncertainty, from "using uncertainty to predict performance" (Fig. 3), to "uncertainty's relationship with caption granularity" (Fig. 5), robust finet-tuning under data sparse regime (Table. 2) and "using uncertainty for outlier detection" (Table 3.). All these different settings provide solid evidence on why we should care about a VLM with uncertainty quantification ability.

- I really like the summarization in Eq. 3 and the text box between Line 165 ! It explains "how AsymVLM works and differs from existing work" in a very intuitive and straightforward way! It's always good to see such analysis explaining the probabilistic objective function from a non-probabilitic perspective.

# Weaknesses

- In section 3.1, the concept of "object" is introduced but I did not quite understand how it is used in the method, it seems to me that the dataset and the method only uses text and image if I understand correctly.

---

> ### Author Rebuttal · Authors · 2025-07-29
>
> Thank you for your thoughtful comments, and encouragement. Please find our responses below.
>
> __Weakness: The concept of an "object"__
> Here "objects" are the unobserved, real‐world entities that both captions and images refer to, e.g., the particular dog (object) that gives rise to one caption ("a dog") and many images (different photos of that same dog).  We introduce $\mathcal{O}$ in Section 3.1 purely as a conceptual bridge to formalize the one‐to‐many (text $\to$ objects $\to$ images) and many‐to‐one (images $\to$ object $\to$ texts) relationships.  In practice, neither our datasets, nor our losses ever see or require explicit object labels: we only train on paired $(t,i)$ tuples.  The "object" layer explains why text is inherently aleatoric-uncertainty-dominant (it abstracts over infinitely many objects) while images, given enough data, admit low aleatoric uncertainty (they pinpoint one object).  The method itself encodes text as a directional distribution and images as points on the representation space $\widetilde{\mathcal{O}}$, without ever instantiating or inferring $o\in \mathcal{O}$.
>
> __Q1: On probabilistic VL encodings and multimodal models__
> Multimodal models like LLaVA typically combine a pre-trained visual encoder (e.g., CLIP’s visual backbone) with a large language model, while leaving the pre-trained textual encoder unused. Thus, AsymVLM is not directly applicable for such multimodal models. However, it is possible to fine-tune LLMs in a similar probabilistic approach as AsymVLM does by adding an additional adaptation layer (adaptor) in the bottleneck layers of LLMs. The adaptor predicts both the mean and concentration parameter of LLM’s text‐embedding distribution. The inverse concentration naturally serves as an uncertainty measure, and and sampling from this distribution during the forward pass promotes diverse generations. Back-propagation is enabled via the reparameterization trick, as is common in VAEs [1], diffusion [2] and flow matching models [3]. Although we have not yet empirically validated this method due to time constraints, it offers a promising avenue for principled uncertainty quantification in foundation models beyond CLIP-style encoders.
>
> __Q2: The adaptor's architecture__
> The architecture of the adaptor is described in the Appendix B.1. The adaptor is implemented with a multi-layer perceptron which consists of two hidden layers, each with 512 dimensions and activated by ReLU function, followed by an output layer that has the same dimensionality as the input with no activation function. Batch normalization is applied to all layers to stabilize training. The un-normalized output of the final layer, denoted as $z^\prime(t)$, is decomposed into $z^\prime(t)=\kappa(t)u(t)$, where the vector $\lVert u(t)\rVert_2=1$ and the scalar $\kappa >0$, serving as the parameters for either vMF or PS distribution.
>
> __Q3: Epistemic v/s aleatoric uncertainties__
> The uncertainty of text embeddings captured by AsymVLM is aleatoric. To estimate the epistemic uncertainty of embeddings, one in general needs to adopt dropout layers or Bayesian neural network in the adaptor, and sample the output with multiple forward passes, which is beyond the scope of our work.
>
> __Q4: Out-of-distribution test images__
> When a test image comes from a domain very different from captions used to train the adapter, our asymmetric uncertainty adapter preserves much more of the pre-trained generalization. As shown in Sec 5.4, Table 2, after training on MS-COCO, FLICKR-30K or CC-200K, the probabilistic CLIP model still achieves strong zero-shot accuracy on CIFAR-10, CIFAR-100 and STL-10 datasets to a large extent. This robustness arises because fine-tuning updates not only the mean $u(t)$ but also the concentration $\kappa(t)$ of each text embedding, by allowing $\kappa(t)$ to decrease (i.e., broaden the distribution) for more abstract captions, AsymVLM avoids over-specialization and remains effective on out-of-distribution images.
>
> __Limitation: Computational costs__
> The overhead of post-hoc adaptation with AsymVLM is negligible compared to pre-training CLIP or SigLIP. This efficiency stems from two factors: (1) the adaptor is a lightweight three-layer MLP operating in the 512-dimensional latent space (for CLIP); (2) all CLIP embeddings for adaptation data are cached after a single forward pass through the pretrained VLM. Adaptation training then proceeds solely on these cached embeddings, avoiding repeated runs through the large model and dramatically reducing compute cost. On an NVIDIA A100, completing 200 epochs of adaptation on either MS-COCO, FLICKR-30K or CC-200k requires under one GPU-hour. Thanks a lot for the suggestion, a discussion on these lines will definitely improve the paper and we will add these details to the camera ready version.
>
>
> ## References
> [1] Kingma DP, Welling M. Auto-encoding variational bayes, 2013 Dec 20
>
> [2] Ho J, Jain A, Abbeel P. Denoising diffusion probabilistic models. Advances in neural information processing systems. 2020;33:6840-51.
>
> [3] Lipman Y, et al. "Flow Matching for Generative Modeling." _The 11th International Conference on Learning Representations_.

---

> ### Author Response · Authors · 2025-08-06
> **Rebuttal feedback**
>
> Hi,
>
> Thanks for the initial review and feedback. As we approach the end of the author-reviewer discussion period, we wanted to request any additional feedback/comments that you may have on the rebuttal we posted earlier.
>
> Thanks,
>
> The Authors

---

### Official Review · Reviewer_qd63 · 2025-07-03

**Clarity:** 2
**Significance:** 2
**Originality:** 3
**Rating:** 4
**Confidence:** 4

**Summary:**

The paper proposes AsymVLM, a post-hoc probabilistic framework for pre-trained vision-language models (VLMs) that captures asymmetric uncertainty in textual and visual embeddings. AsymVLM models text embeddings as probability distributions on the unit hypersphere, while keeping image embeddings deterministic. This design reflects the asymmetric nature of image-text data: textual labels are abstract and ambiguous; images are more concrete but visually varied. Empirical results demonstrate that AsymVLM enhances uncertainty quantification and cross-modal retrieval performance across various datasets, including MS-COCO and Flickr30K.

**Questions:**

1.	Could the authors clarify the semantic meaning and practical implications of the mappings in Table 1 (line 84), especially the one-to-many relationships (e.g., Text -> Object, Object -> Image)?
2.	Why is zero-shot classification only evaluated on small datasets such as CIFAR-10, CIFAR-100, and STL-10? Would the method maintain its claimed robustness on larger-scale benchmarks like ImageNet?
3.	A central design in AsymVLM is to model uncertainty only on the text side while treating image embeddings as deterministic. However, the paper does not provide sufficient justification for this asymmetry. In real-world scenarios--such as medical imaging, satellite imagery, or abstract visual inputs--visual data can also be ambiguous or uncertain. Could the authors clarify why image-side uncertainty is entirely omitted, and whether incorporating it might improve performance or robustness in such settings?
4.	While the framework is evaluated mainly on retrieval and zero-shot classification, can the same asymmetry framework be applied to generative tasks (e.g., image captioning, VQA) where both modalities need rich bidirectional understanding? If so, how would the asymmetry affect generation quality?

**Ethical Concerns:**

["NO or VERY MINOR ethics concerns only"]

**Final Justification:**

The authors' rebuttal has addressed most of my previous concerns. Thus, I decided to raise my score to support this work.

**Limitations:**

Yes.

**Quality:**

2

**Strengths And Weaknesses:**

Strengths:

The paper presents a compelling and original insight by explicitly modeling the asymmetric nature of text-to-image and image-to-text relationships, capturing the fact that textual descriptions are inherently abstract and ambiguous, while images are concrete yet variable. This asymmetry is often overlooked in existing VLM literature, and its incorporation into the design of AsymVLM represents a significant conceptual advancement that deepens our understanding of multimodal alignment.

Weaknesses:
- While the paper presents an informative table outlining the mapping relationships between text, image, object, and representation spaces (line 84-85), it lacks sufficient explanation and justification for several one-to-many assumptions. Key notations such as $\Phi_T$, $\Phi_I$, and their inverses are introduced without thorough semantic grounding or illustrative examples.
- While the paper demonstrates improvements over earlier probabilistic embedding methods such as PFE, PCME++, and ProbVLM, it omits direct comparisons with more recent and competitive approaches like BayesVLM [1] and ProLIP [2]. Although BayesVLM is briefly mentioned and excluded on the grounds that it does not produce scalar embedding uncertainties, this justification is insufficient given that BayesVLM targets the same core problem—uncertainty-aware retrieval—and has shown strong empirical performance. The absence of these baselines weakens the empirical validation and makes it harder to accurately assess the relative strengths of AsymVLM within the current landscape of vision-language uncertainty modeling.
- Zero-shot classification is a critical benchmark for vision-language models, particularly those based on contrastive pretraining like CLIP. AsymVLM evaluates this capability only on three relatively small datasets (CIFAR-10, CIFAR-100, STL-10), which restricts the scope and generalizability of its conclusions. Furthermore, the paper claims that AsymVLM improves robustness under fine-tuning, but fails to convincingly explain why fine-tuning would impair zero-shot performance -- especially given that only a lightweight text-side adapter is trained while the CLIP backbone remains frozen. Since the core parameters of the VLM are untouched, any degradation in zero-shot ability appears inconsistent and unexplained. Without more comprehensive experiments (e.g., on ImageNet) and clearer justification, the claimed preservation of zero-shot capabilities remains insufficiently supported.

---

> ### Author Rebuttal · Authors · 2025-07-29
>
> Thanks for the feedback and suggestions. Please find our responses addressing your concerns and questions below.
>
> __On explanation and justification for several one-to-many assumptions__
> We briefly expand each mapping below, giving concrete examples, and explain why the one-to-many assumptions naturally arise, and motivate our probabilistic treatment of text versus image embeddings.
>
> - Text $\to$ Object $\Phi_T(t)$:
>
> Semantics: $t$ is a text caption (label) while $\Phi_T(t)$ is the set of all real‐world object that fall under the label.
>
> One-to-many: a single word “dog” applies to infinitely many individual dogs in real-world.
>
> - Object $\to$ Image $\Phi_I(o)$:
>
> Semantics: $o$ is a specific object instance while $\Phi_I(o)$ is the set of all images of that same instance.
>
> One-to-many: each specific dog can be imaged under countless viewpoints, lighting conditions, backgrounds, etc.
>
> - Text $\to$ Image $(\Phi_I \circ \Phi_T)(t)$:
>
> Semantics: the set of all images of any object that the label $t$ denotes.
>
> One-to-many: combines the two one-to-many mappings above.
>
> ---
> - Image $\to$ Object $\Phi_I^{-1}(i)$:
>
> Semantics: the (assumed unique) object instance depicted in image $i$.
>
> One-to-one (by assumption): each image is taken of exactly one object instance.
>
> - Object $\to$ Text $\Phi_T^{-1}(o)$:
>
> Semantics: the set of all valid textual descriptions of object $o$.
>
> One-to-many: one object can be described in many different ways (varying specificity, synonyms, context).
>
> - Image $\to$ Text $(\Phi_T^{-1} \circ \Phi_I^{-1})(i)$:
>
> Semantics: all captions that correctly describe the object in image $i$.
>
> One-to-many: inherited from Object$\to$Text, this is mainly because of the abstraction of text data.
>
> ---
> Then with additional Object $\leftrightarrow$ Representation $\psi(o)$ and $\psi^{-1}(\tilde{o})$, which are both one-to-one mapping, it is clear that for CLIP-like VLMs, the text encoder $f_T \approx \psi(o) \circ \Phi_T(t)$ should be one-to-many while the image encoder $f_I \approx \psi(o) \circ \Phi^{-1}_I(i)$ should be one-to-one. Semantically, this corresponds to the irreducible aleatoric uncertainty of language: e.g., no amount of additional examples will resolve the inherent ambiguity of the word "dog." Although images can be uncertain in practice, a well‐trained model with sufficiently diverse visual data can learn to disambiguate them. Any remaining uncertainty is primarily epistemic and thus reducible by collecting more data or improving the model.
>
> __On comparisons to BayesVLM and ProLIP__
> We originally excluded BayesVLM from our baselines because it does not yield scalar uncertainty estimates for embeddings. However, we see that comparisons to ProLIP and BayesVLM add value to the paper. We will add ProLIP and BayesVLM to our baselines and report their performance alongside AsymVLM (see updated results below).
>
> |Dataset|Method|Recall(T2I)|$R^2$(T2I)|$S$(T2I)|Recall(I2T)|$R^2$(I2T)|$S$(I2T)|
> | :---: | :---: | :---: | :---: | :---: | :---: | :---: | :---: |
> |MS-COCO|$\text{ProLIP}$|$0.500\pm0.001$|$0.808\pm0.025$|$-0.908\pm0.031$|$0.304\pm0.002$|$0.876\pm0.007$|$-0.985\pm0.012$|
> | |$\text{AsymVLM}_{\text{vMF}}$|$\mathbf{0.561\pm0.002}$|$\mathbf{0.948\pm0.016}$|$\mathbf{-0.988\pm0.013}$| $\mathbf{0.392\pm0.001}$|$0.984\pm0.004$|$\mathbf{-1.000\pm0.000}$|
> | |$\text{AsymVLM}_{\text{PS}}$|$0.558\pm0.004$|$0.937\pm0.013$|$-0.984\pm0.012$|$0.390\pm0.002$|$\mathbf{0.989\pm  0.001}$|$\mathbf{-1.000\pm0.000}$|
> |FLICKR-30K|$\text{ProLIP}$|$0.678\pm0.000$|$0.829\pm0.032$|$-0.914\pm0.041$|$0.450\pm0.001$|$0.928\pm0.010$|$-0.978 \pm0.005$|
> | |$\text{AsymVLM}_{\text{vMF}}$ | $\mathbf{0.688\pm0.005}$ | $\mathbf{0.860 \pm  0.029}$ | $\mathbf{-0.976\pm0.010}$ | $\mathbf{0.504\pm0.001}$ | $\mathbf{0.960\pm0.006}$ | $\mathbf{-0.995\pm0.006}$ |
> | |$\text{AsymVLM}_{\text{PS}}$|$\mathbf{0.688\pm0.005}$| $0.854\pm0.021$|$-0.947\pm0.032$| $0.498 \pm  0.002$ | $0.946 \pm  0.021$ | $-0.993 \pm  0.006$ |
> | CC-200K | $\text{ProLIP}$ |  $0.351 \pm 0.001$ | $0.968 \pm 0.007$  | $-0.990 \pm 0.009$  |  $0.335 \pm 0.001$ | $0.992 \pm 0.002$  | $\mathbf{-1.000 \pm 0.000}$ |
> | | $\text{AsymVLM}_{\text{vMF}}$ | $\mathbf{0.395 \pm  0.002}$ | $0.990 \pm  0.003$ | $\mathbf{-1.000 \pm 0.000}$ | $\mathbf{0.383 \pm  0.003}$ | $0.991 \pm  0.002$ | $-0.998 \pm 0.005$ |
> | | $\text{AsymVLM}_{\text{PS}}$  | $0.393 \pm 0.001$ | $\mathbf{0.992 \pm 0.004}$ | $\mathbf{-1.000 \pm 0.000}$ | $0.380 \pm 0.002$ | $\mathbf{0.993 \pm 0.001}$ | $\mathbf{-1.000 \pm 0.000}$ |
>
> Due to rebuttal–period time constraints, we were unable to retrain BayesVLM (or re-compute its Hessian) on the same datasets as the other models. Instead, we use the authors’ provided checkpoint. This checkpoint is based on OpenCLIP’s __ViT-B/32-laion2b_s34b_b79k__ foundation model, which was pretrained on a substantially larger dataset than the original CLIP __ViT-B/32__ and thus yields stronger cross-modal retrieval performance [1]. For fairness, we evaluate both BayesVLM and AsymVLM using this identical foundation model as reported below, with BayesVLM’s uncertainties derived from the probabilistic cosine similarity between image–text pairs.
>
> |Dataset|Method|Recall(T2I)|$R^2$(T2I)|$S$(T2I)|Recall(I2T)|$R^2$(I2T)|$S$(I2T)|
> | :---: | :---: | :---: | :---: | :---: | :---: | :---: | :---: |
> | MS-COCO | $\text{BayesVLM}$ | $0.541$ | $0.873$ | $-0.948$ | $0.421$ | $0.913$ | $-0.952$ |
> | | $\text{AsymVLM}_{\text{vMF}}$ | $0.601 \pm 0.003$ | $\mathbf{0.919 \pm 0.012}$ | $-0.985 \pm 0.012$ | $\mathbf{0.421 \pm 0.001}$ | $\mathbf{0.977 \pm 0.002}$ | $\mathbf{-1.000 \pm  0.000}$ |
> | | $\text{AsymVLM}_{\text{PS}}$ | $\mathbf{0.602 \pm 0.003}$ | $0.914 \pm 0.023$ | $\mathbf{-0.990 \pm 0.009}$ | $0.419 \pm 0.001$ | $0.976 \pm 0.003$ | $\mathbf{-1.000 \pm  0.000}$ |
> | FLICKR-30K | $\text{BayesVLM}$ | $0.698$ | $0.778$ | $-0.951$ | $0.517$ | $0.936$ | $-0.987$ |
> | | $\text{AsymVLM}_{\text{vMF}}$ | $\mathbf{0.723 \pm 0.002}$ | $\mathbf{0.884 \pm 0.022}$ | $-0.962 \pm 0.019$ | $\mathbf{0.534 \pm 0.001}$ | $\mathbf{0.979 \pm 0.005}$ | $\mathbf{-1.000 \pm 0.000}$  |
> | | $\text{AsymVLM}_{\text{PS}}$ | $0.721 \pm 0.003$ | $0.871 \pm 0.011$ | $\mathbf{-0.974 \pm 0.015}$ | $0.528 \pm 0.001$ | $0.977 \pm 0.005$ | $-0.999 \pm 0.001$ |
> | CC-200K | $\text{BayesVLM}$ | $0.415$ | $0.914$ | $-0.952$ | $0.397$ | $0.935$ | $-0.987$ |
> | | $\text{AsymVLM}_{\text{vMF}}$ | $0.475 \pm 0.003$ | $\mathbf{0.991 \pm 0.003}$ | $\mathbf{-0.998 \pm 0.005}$ | $\mathbf{0.462 \pm 0.002}$ | $0.984 \pm 0.003$ | $\mathbf{-1.000 \pm 0.000}$ |
> | | $\text{AsymVLM}_{\text{PS}}$ | $\mathbf{0.476 \pm 0.002}$ | $0.987 \pm 0.004$ | $-0.995 \pm 0.006$ | $0.457 \pm 0.002$ | $\mathbf{0.988 \pm 0.006}$ | $-0.998 \pm 0.005$ |
>
> AsymVLM demonstrates higher recall@1 across all three benchmarks and reliable uncertainty estimates with higher $R^2$ and lower $S$ scores. In the camera-ready version, we will include ProLIP and a fully retrained BayesVLM (Hessian-evaluated under identical conditions).
>
> __On zero-shot performance of AsymVLM__
> Fine-tuning a pretrained VLM on a downstream task often comes at the expense of its original zero-shot generality, which is also known as catastrophic forgetting. Empirically, several studies of CLIP-style models have shown that fine-tuning on specialized tasks yields better in-domain accuracy but degrades zero-shot performance on unrelated benchmarks [2]. Here we claim that AsymVLM is a more robust approach to fine-tune VLMs compared to its deterministic counterparts, retaining more generalizability.
>
> Further we report the performance of zero-shot classification of the fine-tuned CLIP model on ImageNet. The experiment results on ImageNet are consistent with the conclusion we had on smaller datasets CIFAR10, CIFAR100 and STL10 in the manuscript. The additional experiment results will be added in the camera-ready manuscript.
>
> | Fine-tuned on | Method | Accuracy |
> | :---: | :---: | :---: |
> | MS-COCO | $\text{Det. AsymVLM}$ | $0.395\pm 0.002$ |
> | | $\text{AsymVLM}_{\text{ps}}$  |   $0.502\pm 0.002$ |
> | | $\text{AsymVLM}_{\text{vMF}}$ | $\mathbf{0.507 \pm  0.003}$ |
> | FLICKR-30K | $\text{Det. AsymVLM}$ | $0.369 \pm 0.007$ |
> | | $\text{AsymVLM}_{\text{ps}}$  | $0.451 \pm  0.003$ |
> | | $\text{AsymVLM}_{\text{vMF}}$ | $\mathbf{0.464 \pm  0.002}$ |
> | CC-200K | $\text{Det. AsymVLM}$     | $0.450 \pm  0.004$ |
> | | $\text{AsymVLM}_{\text{ps}}$  | $0.520 \pm  0.002$ |
> | | $\text{AsymVLM}_{\text{vMF}}$ | $\mathbf{0.527 \pm  0.001}$ |
> | NONE | $\text{CLIP}$ | $0.632$ |
>
> __AsymVLM and generative tasks__
> AsymVLM, as a probabilistic generalization of CLIP and SigLIP, cannot be applied directly to other VLMs. For generative tasks, however, one can extend the same principle by having the text encoder predict not only a mean (or direction) on the hypersphere but also a concentration parameter. The inverse concentration then quantifies uncertainty similarly, and sampling from the resulting vMF or PS distribution during the forward pass encourages diversity in the outputs. Back-propagation through the sampling process can be handled by the reparameterization trick, as routinely done in VAEs [3], diffusion models [4], and flow-matching [5] models. We plan to explore this as future work, and will note this in the camera-ready version.
>
> ## References
> [1] Cherti, M, et al. "Reproducible scaling laws for contrastive language-image learning." _Proceedings of the IEEE/CVF conference on computer vision and pattern recognition_. 2023.
>
> [2] Zhai, Y, et al. "Investigating the catastrophic forgetting in multimodal large language model fine-tuning." _Conference on Parsimony and Learning_. PMLR, 2024.
>
> [3] Kingma, DP., and Max W. "Auto-encoding variational bayes." 20 Dec. 2013.
>
> [4] Ho, J, Ajay J, and Pieter A. "Denoising diffusion probabilistic models." _Advances in neural information processing systems_ 33 (2020): 6840-6851.
>
> [5] Lipman, Y, et al. "Flow Matching for Generative Modeling." _The 11th International Conference on Learning Representations_.

---

> > ### Comment · Reviewer_qd63 · 2025-08-08
> > **Response to authors**
> >
> > Thank you for providing new results and for clarifying most of my concerns. The explanation for modeling uncertainty only on the text side and the discussion on extensions to generative tasks are reasonable, though still theoretical and lacking empirical evidence. Overall, the main issues have been resolved, and the new results improve the completeness of the evaluation. I thus tend to raise the final score to 4.

---

> ### Author Response · Authors · 2025-08-06
> **Rebuttal feedback**
>
> Hi,
>
> Thanks for the initial review. As we approach the end of the author-reviewer discussion period, we wanted to request any feedback/comments that you may have on the rebuttal we posted earlier.
>
> Thanks,
>
> The Authors

---

> ### Author Response · Authors · 2025-08-08
>
> Thank you for the re-evaluation. We’re glad that the additional results addressed most of your concerns and we appreciate your constructive feedback, which helped improve the clarity and completeness of the paper.

---

### Official Review · Reviewer_95zV · 2025-07-03

**Clarity:** 3
**Significance:** 3
**Originality:** 3
**Rating:** 5
**Confidence:** 4

**Summary:**

This paper proposes a method for adapting existing vision-language models (VLMs) into probabilistic models. Instead of maximizing and minimizing the cosine similarity of positive and negative image-text pairs, the authors convert deterministic text embeddings into probability distributions parameterized by the network and then maximize the likelihood the probabilities at the corresponding image embeddings' position. Experimental results demonstrate that the proposed method enables effective fine-tuning with limited data and provides a way to estimate model uncertainty.

**Questions:**

How to understand the relationship between the uncertainty and the equations and setups in Sec. 4.1? For example, does uncertainty means the `variance` in the predicted distribution of text embeddings?

**Ethical Concerns:**

["NO or VERY MINOR ethics concerns only"]

**Final Justification:**

The reviewer has explained some of the minor questions regarding the paper. Hence, I keep my original rating.

**Limitations:**

Yes

**Paper Formatting Concerns:**

No concern on paper formatting

**Quality:**

3

**Strengths And Weaknesses:**

## Strengths
- The motivation is reasonable: a general concept or category described by text can correspond to a wide variety of objects and images in the real world.
- The introduced capabilities—robust fine-tuning with limited data and improved handling of “none-of-the-above” cases—are interesting and practically valuable. These are areas where original models like CLIP often struggle.

## Weaknesses
I don't have major concern on this draft. But I think the asymmetric assumption requires further clarification. The authors claim that textual labels are high-level abstractions, but if the textual labels refer to the labels used in datasets, they can vary significantly in granularity. Some may describe fine-grained objects, while others may be abstract class names, as in ImageNet.  Hence, it would strengthen the paper if the authors discussed the inherent variability of free-form textual data and how their asymmetry assumption accounts for this. In particular, linking this discussion to the uncertainty experiments in Section 5.3 would help ground the assumption in practical evidence and clarify its implications.

---

> ### Author Rebuttal · Authors · 2025-07-29
>
> Thanks a lot for the feedback and encouragement. Please find our response addressing your concerns as follows.
>
> __Clarification on the asymmetric assumption__
> We thank the reviewer for pointing out the wide variability in label granularity, from single‐word class names to richly detailed captions. In AsymVLM this variability is captured by the learned concentration parameter $\kappa$ of the text distribution: Class labels (Level 0 in HierarCap, Sec. 5.3) produce high aleatoric uncertainty with small-valued $\kappa$, whereas more elaborate captions (Levels 1–3) yield progressively lower uncertainty with greater $\kappa$, as shown in Fig. 5 (left panel). However, even the most verbose description cannot convey every pixel‐level detail of an image due to the inherently lossy, highly‐uncertain nature of text data. Thus, our asymmetric assumption both adapts class label to free‐form caption granularity via $\kappa$ and acknowledges the irreducible uncertainty of text data. The monotonic decrease of uncertainty with caption detail in Sec. 5.3 provides concrete empirical grounding for this design choice. We agree that explicit discussion over various label granularity helps ground the assumption and will add it in the camera-ready manuscript upon acceptance.
>
> __Understanding the relationship between the uncertainty, equations and setups in Sec. 4.1__
> We quantify text‐embedding uncertainty as  $u(t)=\frac{1}{\kappa(t)}$, due to the fact that for both the vMF and PS distributions on $\mathbb{S}^{d-1}$, the (angular) variance $\sigma^2$ is a strictly decreasing function of $\kappa$ [1, 2]. Equivalently, larger $\kappa$ concentrates mass more tightly around the mean direction (lower $\sigma^2$), so $1/\kappa$ grows monotonically with the distribution’s variance, providing a natural scalar value for uncertainty estimate for probabilistic embeddings.
>
> ## References
> [1] De Cao, Nicola, and Wilker Aziz. "The power spherical distribution." arXiv preprint arXiv:2006.04437 (2020).
>
> [2] Sra, Suvrit. "A short note on parameter approximation for von Mises-Fisher distributions: and a fast implementation of I s (x)." Computational Statistics 27.1 (2012): 177-190.

---

> > ### Comment · Reviewer_95zV · 2025-08-08
> >
> > Thank the authors for the further explaining those details and concept. I will maintain my original rating.

---

> > > ### Author Response · Authors · 2025-08-08
> > >
> > > We thank you again for the detailed comments in the review and for your efforts in addressing our rebuttal. Your feedback has been invaluable in improving the manuscript.

---

> ### Author Response · Authors · 2025-08-06
> **Rebuttal feedback**
>
> Hi,
>
> Thanks for the initial review. As we approach the end of the author-reviewer discussion period, we wanted to request any feedback/comments that you may have on the rebuttal we posted earlier.
>
> Thanks,
>
> The Authors

---

### Official Review · Reviewer_Hj1D · 2025-07-03

**Clarity:** 3
**Significance:** 3
**Originality:** 3
**Rating:** 4
**Confidence:** 3

**Summary:**

The paper discusses that while VLMs perform well, they fail to capture ambiguity and uncertainty in the data. There are existing works handling this with post-hoc methods. However, these methods do not handle the asymmetric uncertainty. The paper claims that having the representations on a hypersphere results in suboptimal performance. The paper proposes AsymVLM, which builds probabilistic embeddings from pre-trained VLMs on the hypersphere, enabling uncertainty quantification.

**Questions:**

Please check weaknesses.

**Ethical Concerns:**

["NO or VERY MINOR ethics concerns only"]

**Final Justification:**

The rebuttal has answered my questions so I change my rating from 3 to 4.

**Limitations:**

Weaknesses are addressed.

**Paper Formatting Concerns:**

No formatting concerns

**Quality:**

3

**Strengths And Weaknesses:**

# Strength:

The paper dives into the asymmetric nature of text and image uncertainty. The motivation is valid and clear. The problem formulation is novel and intuitive.

The paper provides ablation studies investigating uncertainty levels. The paper also provides experiments comparing with several models, showing better performance of the model. In addition, there are various applications, including zero-shot and none-of-the-above.

The writing is without grammar or spelling errors.

# Weaknesses:

## Method explanation:

While the writing in general is good, the overall method explanation, task definition and experiments need polishing to be better grasped.

Lines 30–33 discuss the asymmetric nature of text and images, but the explanation is difficult to follow. Referencing Figure 1 and including concrete examples could improve clarity. Since this section appears in the introduction and addresses the core motivation of the paper, the explanation must be clear and compelling.

What’s the numerical representation space in line 84?

It would also be beneficial to give an overview of the building VLM section with examples to make it easy to grasp.

What is the role of the text adaptor introduced in 4.1?

The introduction misses an overview of the contributions of the proposed method and a distinction from the existing works.

## Uncertainty:

The paper makes statements or assumptions about the uncertainty; however, it still needs to provide support for the claims and assumptions. For example, I have a problem with the simplification and generalisation in lines 104-110. Can the paper provide support and references for the explanation of uncertainty?

In Figure 3, how are different uncertainty levels generated or calculated? What is the measure? What is the data? Line 186 references Figure 3; however, it is not clear what the setup is. The same stands for the experiment setups in 5.3.

Line 49: Why do the models fail to capture the inherent ambiguity? Are there any supports for this? Any references investigating?


Small writing comments:

In line 79, is there a space in the middle of e.g.?

Figure 1 is not referenced nor explained in the text.

---

> ### Author Rebuttal · Authors · 2025-07-29
>
> Thank you for the helpful feedback. Please find our responses below.
>
> __Improving method explanation__
> In the camera-ready manuscript, we will address the concerns by clarifying the problem formulation and method motivation in Section 3, and by adding an additional subsection about the calculation of the uncertainty from the resulting probabilistic embedding in Section 4, to eliminate any potential ambiguities.
>
> __On the asymmetric nature of text and images__
> Crucially, text and images exhibit different kinds of one-to-many ambiguity (Fig. 1, left). The word "dog" can denote any of infinitely many dog instances (objects), each of which can in turn be photographed under many viewpoints, lighting conditions, etc. Thus text $\to$ object is one-to-many via abstraction, and object $\to$ image is one-to-many via visual variability. In contrast, a photo of a golden retriever unambiguously depicts one physical object but admits multiple valid captions (e.g., "a dog," "a golden retriever," "a pet", "an animal"). Thus image$\to$text is one-to-many through diverse linguistic descriptions. Existing post-hoc probabilistic adapters treat these two retrieval directions symmetrically.
>
> Additionally, existing methods overlook the fact that embeddings of VLMs pre-trained with cosine similarity lie on the unit hypersphere. We argue that modeling this asymmetry by capturing the uncertainty for text only all on the sphere, is the key to better cross-modal uncertainty estimation.
>
> __The numerical representation space__
> Numerical representation space: The numerical representation space (Line 84) is the joint embedding space into which both image and text inputs are projected by the encoders of a VLM. In this space, semantically matching images and texts yield embeddings that are close under a chosen similarity metric. The numerical representation space should be $\mathbb{R}^d$ for Euclidean distance as similarity metric, and $\mathbb{S}^{d-1}$ for cosine similarity, where $d$ is the dimension of the embedding.
>
> __Overview of building VLMs__
> Existing datasets such as MS-COCO collect data in the form of $\{(t_n, i_n)\}_{n=1}^N$ where textual caption $t_n$ is able to describe image $i_n$. Introducing the (real-world) object space $\mathcal{O}$, Text$\to$Object mapping $\Phi_T$ and Object$\to$Image mapping $\Phi_I$, the relationship between $t_n$ and $i_n$ can be be formally stated as: there exist an object $o_n\in\mathcal{O}$ such that $t_n$ can be used to describe $o_n$, i.e., $o_n \in \Phi_T(t_n)$, and $i_n$ is an image of $o_n$, i.e. $i_n \in \Phi_I(o_n)$. However, with no concrete way to represent real-world object space, we assume that there exist a numerical representation space $\widetilde{\mathcal{O}}$, such that for any object $o_n\in\mathcal{O}$, there exist a unique real-valued vector $\tilde{o_n}\in\widetilde{\mathcal{O}}$. The Object $\leftrightarrow$ Representation mapping is denoted as $\psi:\mathcal{O}\to\widetilde{\mathcal{O}}$ and $\psi^{-1}:\widetilde{\mathcal{O}}\to\mathcal{O}$.
>
> For CLIP-like VLMs, we aim to build two encoder to learn the mappings from images and texts to the joint numerical representation space, i.e. Image $\to$ Representation and Text$\to$Representation. The mappings can be decompose as Image $\to$ Object $\to$ Representation, i.e., $\psi \circ \Phi_I^{-1}$ and Text $\to$ Object $\to$ Representation, i.e., $\psi \circ \Phi_T$. Thus, with finite samples $\{(t_n, i_n)\}_{n=1}^N$, building VLMs can be interpreted as $f_I \approx \psi\circ \Phi_I^{-1}$ and $f_T \approx \psi \circ \Phi_T$, where $f_I$ and $f_T$ are image and text encoders respectively.
>
> __Role of the text adaptor__
> We convert deterministic text embeddings into probabilistic ones by feeding them through the AsymVLM adaptor (a small neural network) that predicts the parameters of our chosen spherical distribution (von Mises–Fisher or power spherical).
>
> __On clearly mentioning contributions__
> To make both our contributions and our differences to prior work explicit, we will add the following paragraph at the end of Introduction. Thanks a lot for the suggestion.
>
> "Contributions:  This paper makes three main contributions:
> - We formally identify and address the asymmetric uncertainty structure inherent in vision–language data, high aleatoric uncertainty in text (one-to‐many mapping) versus lower (aleatoric) uncertainty in images.
> - We propose AsymVLM, an adapter that exploits the asymmetric uncertainty structure and performs post-hoc probabilistic adaptation on the unit hypersphere, rather than in Euclidean space. We also show that AsymVLM is a natural extension of CLIP’s cosine-similarity loss with the added ability to quantify uncertainty.
> - Empirically, AsymVLM yields more accurate uncertainty estimates and higher cross-modal retrieval accuracy on MS-COCO, Flickr30K and CC-200k benchmarks.  We further demonstrate its advantages in robust fine-tuning, zero-shot classification and "none-of-the-above" rejection.  "
>
> __Clarifications about the uncertainty__
>
> 1. __Clarification for Line 104-110:__ From an uncertainty-quantification perspective we adopt the standard taxonomy of aleatoric vs. epistemic uncertainty [1, 2]. Aleatoric uncertainty captures irreducible data ambiguity, e.g. the caption "a dog" denotes an entire class of breeds, poses and contexts, so no amount of additional training can pinpoint a single underlying object. Epistemic uncertainty captures model ignorance that could be reduced with more or better-distributed data, e.g., images, while depicting a unique scene with low aleatoric spread, may still suffer from coverage gaps if certain viewpoints or lighting conditions were under-represented during training. This asymmetry yields inherently broad aleatoric uncertainty in text versus more data‐dependent epistemic uncertainty in images. In practice, aleatoric uncertainty is modeled via parameterized distributions, whereas epistemic uncertainty is estimated by Monte Carlo methods (e.g., sampling from dropout layers or Bayesian neural networks). Accordingly, in this work, focusing on aleatoric uncertainty, we represent text embeddings as directional distributions on the unit hypersphere and retain deterministic image vectors.
>
> 2. __How are different uncertainty levels generated or calculated?__
> The uncertainty of each text embedding is quantified by the inverse concentration parameter of the embedding distribution, $1/\kappa(t)$. Following the probVLM [3] convention, we sort all the uncertainty estimates in ascending order and split them into ten equally sized bins (levels 0–9), so that larger uncertainty values are assigned to higher levels. This quantile‐based scheme lets us evaluate and compare the quality of uncertainty estimates from different methods by observing that cross‐modal retrieval accuracy declines monotonically as the uncertainty level increases.
>
> 3. __Why do the models fail to capture the inherent ambiguity?__
> As noted in Section 3.1, because the mapping Text$\to$Object is inherently one‐to‐many, the induced Text$\to$Representation should likewise be one‐to‐many.  In practice, however, VLM text encoders approximate this mapping deterministically, i.e., the text encoder $f_t$ maps each text to a single point in the representation space, thereby failing to capture textual ambiguity. Prior works [3,4,5] have shown that probabilistic embeddings, which represent each text as a distribution rather than a point, more faithfully model semantic uncertainty and yield improvements on downstream tasks. Our empirical results further validate this conclusion.
>
> __Small writing issues__
> We apologize for the identified writing issues and will fix them in the camera-ready version. In particular, we will properly reference Figure 1 in Section 3.1 and refine the surrounding discussion to clarify the paper’s main idea.
>
> ## References
> [1] Hüllermeier, Eyke, and Willem Waegeman. "Aleatoric and epistemic uncertainty in machine learning: An introduction to concepts and methods." _Machine learning_ 110.3 (2021): 457-506.
>
> [2] Gawlikowski, Jakob, et al. "A survey of uncertainty in deep neural networks." _Artificial Intelligence Review_ 56.Suppl 1 (2023): 1513-1589.
>
> [3] Upadhyay, Uddeshya, et al. "Probvlm: Probabilistic adapter for frozen vison-language models." _Proceedings of the IEEE/CVF International Conference on Computer Vision_. 2023.
>
> [4] Chun, Sanghyuk. "Improved Probabilistic Image-Text Representations." _The Twelfth International Conference on Learning Representations_. 2024.
>
> [5] Shi, Yichun, and Anil K. Jain. "Probabilistic face embeddings." _Proceedings of the IEEE/CVF international conference on computer vision_. 2019.

---

> > ### Comment · Reviewer_Hj1D · 2025-08-04
> > **reply to the rebuttal**
> >
> > I thank the authors for providing a good rebuttal.
> > I have a question wrt to the failure of other models.
> > Could the authors provide more explanations on how and where the mentioned references discuss the failure? It would be beneficial to answer my question. Thanks.

---

> ### Author Response · Authors · 2025-08-04
>
> Thanks for the follow-up question! We realized that your question can be read in two different ways, and we therefore address both possibilities below.
>
> - If by "other methods" you refer to deterministic VLMs (e.g., CLIP, BLIP, SigLip): These methods map every caption to a single point, so they cannot represent the inherent one-to-many relations in textual information. This limitation, often framed as an inability to capture aleatoric uncertainty, has been analyzed in Section 2 (Related work for probabilistic embeddings) in PCME [1], Section 3 (Limitations of Deterministic Embeddings) in PFE [2], Section 2 (Related work for vision-language models) in ProbVLM [3] and Section 2.2 (Stochastic embedding) in HIB [4].
>
> - If "other models" refers to existing post-hoc probabilistic adapters: Existing post-hoc probabilistic adaptation methods model text and image uncertainties symmetrically and assume Euclidean geometry, while pre-trained VLM embeddings lie on the unit hypersphere and exhibit modality-specific uncertainty patterns. Both our theoretical analysis and the ablation study in Fig. 4 show that accounting for both (I) asymmetric uncertainty and (II) unit hyperspherical geometry is essential, removing either component degrades recall and weakens the uncertainty estimation.
>
> We will update line 49 to be more clear and specific as you outlined:
> "However, deterministic pretrained VLMs may not fully capture the ambiguity [3, 4] arising from one-to-many relationships involving textual and image inputs, motivating the need for probabilistic/stochastic embeddings [3, 4]"
>
> We regret the confusion caused and thank you for the suggestion.
>
> ## References
> [1] Chun, Sanghyuk, et al. "Probabilistic embeddings for cross-modal retrieval." _Proceedings of the IEEE/CVF conference on computer vision and pattern recognition_. 2021.
>
> [2] Shi, Yichun, and Anil K. Jain. "Probabilistic face embeddings." _Proceedings of the IEEE/CVF international conference on computer vision_. 2019.
>
> [3] Upadhyay, Uddeshya, et al. "Probvlm: Probabilistic adapter for frozen vison-language models." _Proceedings of the IEEE/CVF International Conference on Computer Vision_. 2023.
>
> [4] Oh, Seong Joon, et al. "Modeling Uncertainty with Hedged Instance Embeddings." International Conference on Learning Representations.

---

> > ### Author Response · Authors · 2025-08-07
> >
> > Hi,
> >
> > Thanks again for your engagement with us, and for your valuable feedback towards improving the paper. We appreciate your earlier comments, and would be happy to clarify/address any remaining feedback you might have, as we approach the end of the interaction period.
> >
> > Best regards,
> >
> > The Authors

---

### Author Response · Authors · 2025-08-08
**Summary of the promised changes to the paper**

Dear Reviewers and Area Chair(s),

We very much appreciate the insightful comments and discussions. Below is a consolidated summary post-rebuttal, listing the clarifications, additional experiments, and promised paper changes reported in the individual rebuttals.

1. Core idea and contributions
- Vision-language data have intrinsic asymmetric uncertainty structure: text exhibits high, irreducible (aleatoric) ambiguity, whereas images mainly suffer from reducible (epistemic) uncertainty. We formalize this via a latent object space $\mathcal{O}$. Text $\to$ Object and Object $\to$ image are both one-to-many, while image $\to$ Object is (by assumption) one-to-one. Therefore text needs probabilistic representations while images can remain deterministic.
- AsymVLM is a lightweight post-hoc adapter that maps every CLIP (or SigLIP) text embedding $t$ to a directional distribution (vMF or Power-Spherical) on the unit hypersphere, parameterized by mean direction $\mu(t)$ and concentration $\kappa(t)$. Uncertainty $u(t)$ is quantified by $1/\kappa(t)$
- Empirically, AsymVLM improves both recall@1 and calibration on MS-COCO, Flickr30K and CC-200K, and yields more robust fine-tuning (higher zero-shot classification accuracy) than deterministic counterparts.

2. Methodological clarifications to be added in the camera-ready version upon acceptance
- Expanded explanation of the object and representation space (Sec. 3.1), the adaptor architecture (Sec. B.1), and the derivation of the uncertainty measure $u(t)$ (Sec. 4).
- An explicit paragraph in the introduction stating the three contributions above.
- A new subsection in Sec. 4 detailing how variance on is a monotonically decreasing function of $\kappa$ for both vMF and PS, justifying the uncertainty measure $u(t)$.
- Discussion of label-granularity experiments (Sec. 5.3) showing monotonic $\kappa$ growth from class names to full captions.
- Clear differentiation between aleatoric (modeled by $\kappa(t)$) and epistemic (would require Monte-Carlo sampling, out of scope here) uncertainty.

3. New experimental results in the paper, as suggested by reviewers
- Added comparisons to ProLIP [1] and BayesVLM [2]. AsymVLM outperforms both in recall@1 and in uncertainty metrics (higher $R^2$, lower $S$).
- Included zero-shot accuracy of the fine-tuned models on ImageNet and the proposed AsymVLM outperforms its deterministic counterparts.
- Reported negligible compute overhead: one GPU-hour on an A100 for 200 epochs as all CLIP/SigLIP features are cached.

4. Minor revisions scheduled
- We will address all identified writing issues (missing Figure 1 reference, typos, clearer section cross-links).

Clarifications on the scope and future work: 1). While the current adapter targets CLIP-style encoders, the same principle can be extended to generative or LLM-based multi-modal models by predicting $(\mu, \kappa)$ at their text bottlenecks, which can be sampled and back propagated with the re-parameterization trick [3].  2). Epistemic uncertainty can be layered on top of AsymVLM via dropout/Bayesian adapters and this is left for future study.

We hope this unified response clarifies the motivation, theoretical grounding, and empirical strength of AsymVLM, and addresses all open questions raised during the review and discussion period. We thank the committee again for the thoughtful feedback and believe the planned revisions will substantially strengthen the paper.

Best regards,
The Authors

---

### References
[1] Chun, Sanghyuk, et al. "Probabilistic Language-Image Pre-Training." _The Thirteenth International Conference on Learning Representations_.

[2] Baumann, Anton, et al. "Post-hoc probabilistic vision-language models." _arXiv preprint arXiv:2412.06014_ (2024).

[3] Kingma, DP., and Max W. "Auto-encoding variational bayes." 20 Dec. 2013.

---

### Note · Authors · 2025-08-11

Dear Reviewers and Area Chair(s),


Please see the 'Official Comment' right below, listing the summary of proposed changes and final clarifications.


Best regards,

The Authors

---

### Decision · Program_Chairs · 2025-09-17

**Decision:**

Accept (poster)

**Comment:**

The paper introduces AsymVLM, a lightweight, post-hoc framework to equip pre-trained Vision-Language Models (VLMs) with uncertainty quantification capabilities. The central scientific claim is that vision-language data possesses an intrinsic asymmetric uncertainty structure: text is subject to high, irreducible (aleatoric) ambiguity, whereas visual data has primarily reducible (epistemic) uncertainty. AsymVLM operationalizes this insight by modeling text embeddings as probabilistic distributions on the unit hypersphere, while keeping image embeddings deterministic. The findings demonstrate that this asymmetric approach improves cross-modal retrieval, provides better-calibrated uncertainty estimates, and enhances the robustness of fine-tuning on downstream zero-shot classification tasks.

The reviewers find the paper's core idea novel and well-motivated ("The paper presents a compelling and original insight by explicitly modeling the asymmetric nature of text-to-image and image-to-text relationships... This asymmetry is often overlooked in existing VLM literature, and its incorporation into the design of AsymVLM represents a significant conceptual advancement." by qd63), the empirical evaluation strong ("The empirical results are very thorough and interesting! The experiment considers a wide range of settings related to uncertainty... All these different settings provide solid evidence on why we should care about a VLM with uncertainty quantification ability" by X6GY), and the paper well written ("The paper is very well organized and well written, and the figures are clear, illustrative, and informative." by X6GY).

In the meantime, reviewers also mentioned that the core asymmetry assumption and the methodology needs more justification ("it would strengthen the paper if the authors discussed the inherent variability of free-form textual data and how their asymmetry assumption accounts for this." by 95zV, and "While the writing in general is good, the overall method explanation, task definition and experiments need polishing to be better grasped." by Hj1D), and the experimental validation can be further improved ("omits direct comparisons with more recent and competitive approaches like BayesVLM and ProLIP" and that the zero-shot evaluation "restricts the scope and generalizability of its conclusions" by qd63).

During rebuttal period, the paper's initial weaknesses, primarily the omission of key recent baselines and the lack of large-scale zero-shot experiments, were decisively addressed during the rebuttal period. The authors provided significant new results comparing their method against state-of-the-art models and expanded their evaluation to ImageNet, substantially strengthening the paper's claims and leading reviewers to raise their scores. Therefore, the recommendation for a acceptance is based on the paper's elegant and original contribution, now supported by a rigorous and more complete empirical validation that resolves all major initial concerns, which passes NeurIPS' high quality bar.